# Antimicrobial mitochondrial reactive oxygen species induction by lung epithelial immunometabolic modulation

Yongxing Wang[1], Vikram V. Kulkarni[1,2], Jezreel Pantaleón García[1], Miguel M. Leiva-Juárez[1], David L. Goldblatt[1], Fahad Gulraiz[1], Lisandra Vila Ellis[1], Jichao Chen[1], Michael K. Longmire[1,2], Sri Ramya Donepudi[3], Philip L. Lorenzi[3], Hao Wang[4], Lee-Jun Wong[4†], Michael J. Tuvim[1], Scott E. Evans[1,2]*

1 Department of Pulmonary Medicine, University of Texas MD Anderson Cancer Center, Houston, Texas, United States of America, 2 University of Texas MD Anderson Cancer Center UTHealth Graduate School of Biomedical Sciences, Houston, Texas, United States of America, 3 Department of Bioinformatics and Computational Biology, University of Texas MD Anderson Cancer Center, Houston, Texas, United States of America, 4 Molecular and Human Genetics, Baylor College of Medicine, Houston, Texas, United States of America

† Deceased.
* seevans@mdanderson.org

## Abstract

Pneumonia is a worldwide threat, making discovery of novel means to combat lower respiratory tract infection an urgent need. Manipulating the lungs' intrinsic host defenses by therapeutic delivery of certain pathogen-associated molecular patterns protects mice against pneumonia in a reactive oxygen species (ROS)-dependent manner. Here we show that antimicrobial ROS are induced from lung epithelial cells by interactions of CpG oligodeoxynucleotides (ODN) with mitochondrial voltage-dependent anion channel 1 (VDAC1). The ODN-VDAC1 interaction alters cellular ATP/ADP/AMP localization, increases delivery of electrons to the electron transport chain (ETC), increases mitochondrial membrane potential ($\Delta_{\Psi_m}$), differentially modulates ETC complex activities and consequently results in leak of electrons from ETC complex III and superoxide formation. The ODN-induced mitochondrial ROS yield protective antibacterial effects. Together, these studies identify a therapeutic metabolic manipulation strategy to broadly protect against pneumonia without reliance on antibiotics.

## Author summary

To protect broadly against pneumonia, CpG oligodeoxynucleotides promote mitochondrial immunometabolic modulation that results in antimicrobial mtROS generation in lung epithelium.

---

**Data Availability Statement:** The datasets generated for the Reverse-phase protein array data is available in the Evans Laboratory GitHub repository (www.github.com/evanslaboratory/Datasource).

**Funding:** This work is supported by U.S. National Institutes of Health grants R01 HL117976, DP2 HL123229, and R35 HL144805 to S.E.E. and grants S10 OD012304-01 and P30 CA016672 to the Metabolomics Core Facility. The funders had no role in study design, data collection and analysis, decision to publish, or preparation of the manuscript.

**Competing interests:** I have read the journal's policy and the authors of this manuscript have the following competing interests: MJT and SEE are authors on U.S. patent 8,883,174, "Stimulation of Innate Resistance of the Lungs to Infection with Synthetic Ligands." MJT and SEE own stock in Pulmotect, Inc. All other authors declare that no conflicts of interest exist.

## Introduction

Pneumonia has long been recognized as a leading cause of death among healthy and immuno-suppressed people worldwide [1–3]. Pneumonia management has historically focused on patient-extrinsic factors, such as antibiotic administration [4,5]. To address such challenges as increasing antibiotic resistance and newly emerging infections, our laboratory focuses on manipulating vulnerable patients' intrinsic antimicrobial defenses to broadly protect them against pneumonia. We advance a strategy of activating the lungs' mucosal defenses to induce broad, pathogen-agnostic protection via airway delivery of synthetic Toll-like receptor (TLR) agonists.

Once regarded as simple airflow conduits or inert barriers, the airway and alveolar epithelia are critical immune effector cells that supplement the lungs' mucosal immune defenses by undergoing fundamental structural and functional changes upon encountering pathogens [6–8]. These cells sense pathogens via pattern recognition receptors (PRRs), modulate lung leukocyte responses through cytokine and chemokine expression, and release microbicidal molecules such as reactive oxygen species (ROS) and antimicrobial polypeptides (AMPs) [9–11]. Harnessing this defensive immune function, we developed a protective PRR agonist therapeutic comprised of a synthetic diacylated lipopetide ligand for TLR2/6 (Pam2CSK4, "Pam2") and a class C unmethylated CpG oligodeoxynucleotide ligand for TLR9 (ODN M362, "ODN"). A single prophylactic treatment with this non-intuitive dyad of ligands ("Pam2-ODN") for spatially segregated TLRs yields substantial protection against pneumonia [12–16].

We recently reported that Pam2-ODN-induced antimicrobial protection requires therapeutic induction of ROS from both mitochondrial and dual oxidase sources [17,18], but the molecular mechanisms responsible for inducible antimicrobial ROS generation remained unresolved. Here, we find that ODN induces mitochondrial ROS (mtROS) production via metabolic modulation that alters mitochondrial electron transport chain (ETC) activity in a mitochondrial membrane potential ($\Delta_{\Psi_m}$)-dependent manner. These findings provide novel insights into development of metabolic strategies to protect against otherwise lethal pneumonias in vulnerable populations.

## Results

### Induction of epithelial mtROS by CpG ODN

Having reported that Pam2 and ODN M362 are both required for maximal antimicrobial protection in a manner that depends on inducible ROS production from both mitochondria and dual oxidases, we sought to determine which ligand(s) induce mtROS production. As shown in Fig 1A, ODN M362 alone induced as much mtROS generation as the Pam2-ODN combination from human lung epithelial (HBEC3-KT) cells, revealing ODN M362 as the main driver of this response. This capacity to induce mtROS was not common to all nucleic acid treatments (Fig 1A), but was observed following treatment of HBEC-3KT cells with various types of oligodeoxynucleotides (Fig 1B). Similarly, ODN M362 induced mtROS in mouse lung epithelial (MLE-15) cells (Fig 1C), as well as primary human and mouse lung epithelial cells (Fig 1D–1E), regardless of which mtROS detector was used (S1 Fig). HBEC3-KT cells and MLE-15 cells are the principal models presented throughout this work, because these two cell types can represent a wide range of epithelia (human cells of bronchial origin and mouse cells with more alveolar characteristics, respectively). This is potentially important, because the effect of mtROS generation upon first encountering pathogens in the airways may be different than interactions as pneumonic infections progress to the parenchyma. However, it is notable that robust induction of mtROS was observed from all tested types of lung epithelial cells

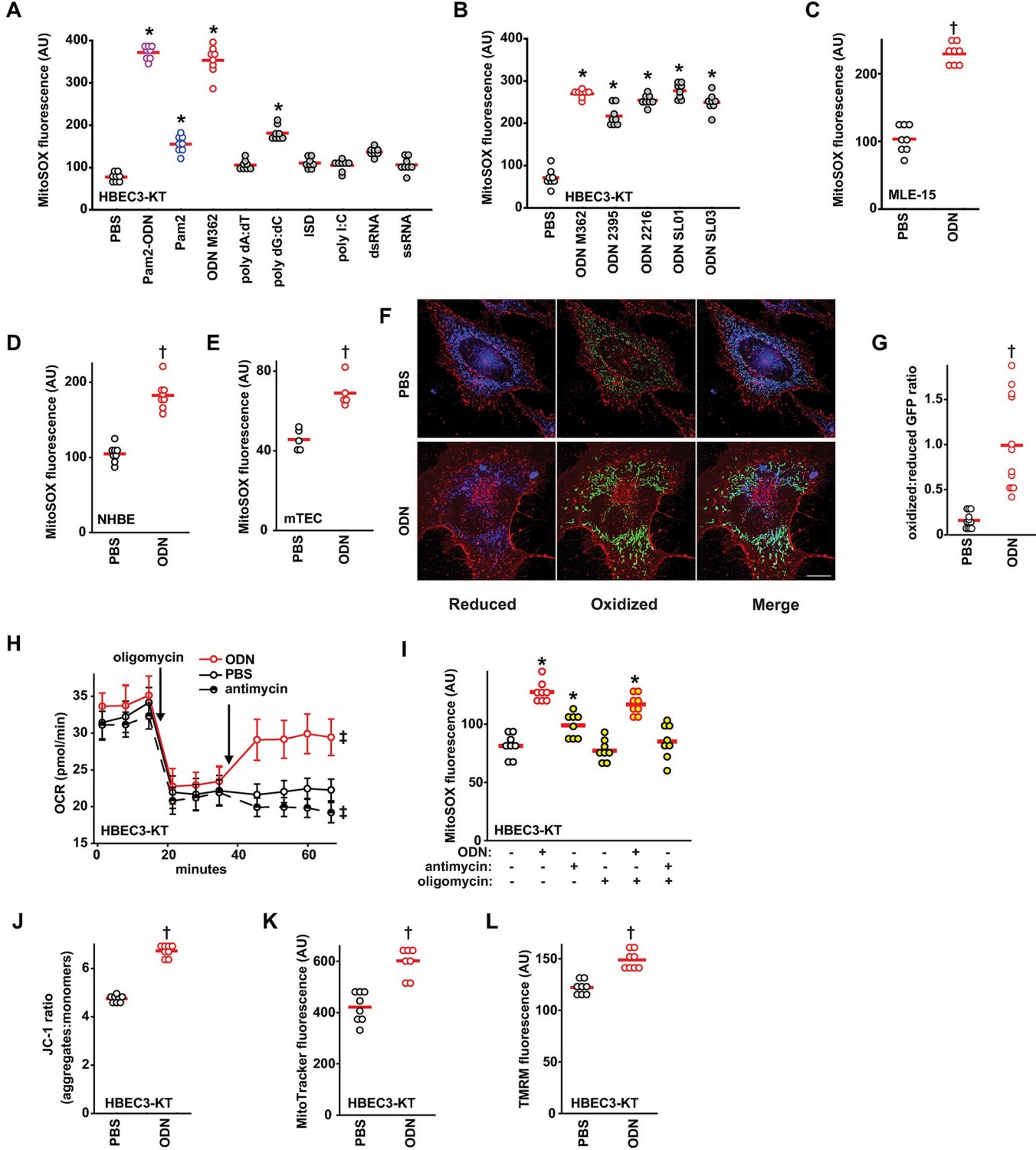

**Fig 1. Induction of epithelial mtROS by CpG ODN.** (**A**) mtROS production from HBEC3-KT cells after treatment with pathogen associated molecular patterns. (**B**) mtROS production from HBEC3-KT cells after treatment with the indicated ODNs. mtROS production after treatment with ODN from mouse lung epithelial cell lines (**C**) and primary human (**D**) and primary mouse (**E**) lung epithelial cells. (**F**) Representative fluorescence images primary tracheal epithelial cells harvested from mt-roGFP mice treated with PBS or ODN. Images shown as gradient of color intensity from the reduced (blue) form to the oxidized (green) form of roGFP. Scale bar, 50 μm. (**G**) Ratio of the fluorescence intensity of the oxidized:reduced roGFP from **F,** quantified at 488 nm and 405 nm, respectively. (**H**) Oxygen consumption following the indicated treatment by Seahorse XFe96 Flux Analyzer, shown as mean ± SEM. (**I**) ODN-induced mtROS production from HBEC3-KT cells in the presence of the indicated inhibitors. Mitochondrial membrane potential $\Delta_{\Psi m}$ measurement in HBEC3-KT

cells after ODN treatment assessed by JC-1 (**J**), MitoTracker (**K**) and TMRM (**L**). * p<0.001 vs. PBS by one-way ANOVA using Holm-Sidak method, except A which use Tukey method due to failed normality testing; † p<0.001 vs PBS by two-way Student's t test; ‡ p <0.008 vs PBS treated by one-way ANOVA using Dunnett's test for multiple comparisons. ODN, oligodeoxynucleotide; ISD, immune stimulating DNA; mTEC, primary mouse tracheal epithelial cells; NHBE, primary normal human bronchial epithelial cells; GFP, green fluorescent protein; OCR, oxygen consumption rate; TMRM, tetramethylrhodamine.

whether mouse or human, whether the cells were primarily harvested or immortalized, and whether they were of airway or alveolar origin (S2 Fig) Fluorescence microscopy of primary lung epithelial cells from mice expressing redox-sensitive mitochondrial GFP (mt-roGFP) [19, 20] revealed that, at baseline, mitochondria display a predominantly reduced GFP phenotype, whereas treatment with ODN induces a predominantly oxidized mitochondrial phenotype (Fig 1F–1G). Under normal conditions, oxidative phosphorylation consumes >95% of cellular oxygen [21], but mtROS formation also requires free oxygen [22]. By inhibiting oxidative phosphorylation with oligomycin, we can then demonstrate increased oxygen consumption by superoxide ($O_2 \bullet^-$) production following ODN (Fig 1H), which is consistent with the observation of increased mitoSOX fluorescence (Fig 1I). This ODN-induced mtROS production is consistently associated with increased $\Delta \Psi_m$, as assessed by multiple assays (Fig 1J–1L).

## CpG ODN alters electron transport chain activity and energy production

mtROS production is tightly regulated by electron transport chain (ETC) activity [23]. To understand the mechanisms of ODN-induced mtROS production, we analyzed the enzymatic activity of the ETC complexes in HBEC3-KT cells, finding that ODN treatment induces a 35% increase in complex II activity and an 82% decrease in complex III activity, along with an increase in citrate synthase activity (Figs 2A and S3). We also found modest but statistically significant reductions in complex V activity, suggesting ODN may interfere with mitochondrial energy production (S3 Fig). The change in ETC complex activity was not accompanied by changes in mitochondrial protein concentrations (Figs 2B and S4), suggesting that the ODN effect is mediated by manipulating ETC function rather than altering mitochondrial mass.

Since the major energy output of ETC activity is ATP, we investigated the impact of ODN treatment on cellular ATP levels. ODN treatment caused a rapid decline in whole-cell ATP concentrations with a nadir around 30 min that recovered by 90 min (Fig 2C). In contrast, cellular ADP and AMP levels persistently rose following ODN exposure (Fig 2D–2E). These effects on ATP, ADP and AMP were ODN dose-dependent (S5 Fig). A drop-and-recovery pattern similar to that of ATP was seen in cellular NADH levels after ODN treatment, whereas NADPH levels were largely unaffected by ODN treatment (Figs 2F–2G and S5). NAD/NADH are electron receptor and donor that links large molecule catabolism to mitochondrial energy production. The congruent temporal patterns of NADH and ATP support a hypothesis that ODN stimulates catabolic reactions related to mitochondrial energy production. In contrast, NADPH is primarily produced in the anabolic pentose phosphate pathway, which ODN does not appear to perturb. Levels of reduced glutathione persistently declined after ODN treatment (Fig 2H), consistent with the continuing production of ODN-induced ROS.

## CpG ODN blocks mitochondrial nucleotide transition

As TLR9 is the established intracellular sensor for CpG ODNs [24], we examined whether TLR9 activation regulates ODN-induced mtROS generation. To our surprise, ODN treatment still fully induced mtROS production in primary mouse lung epithelial cells isolated from *Tlr9* knockout mice or from mice lacking downstream TLR signaling molecules MyD88 or TRAF6 (S6 Fig), indicating that ODN-induced mtROS generation does not require TLR9 signaling.

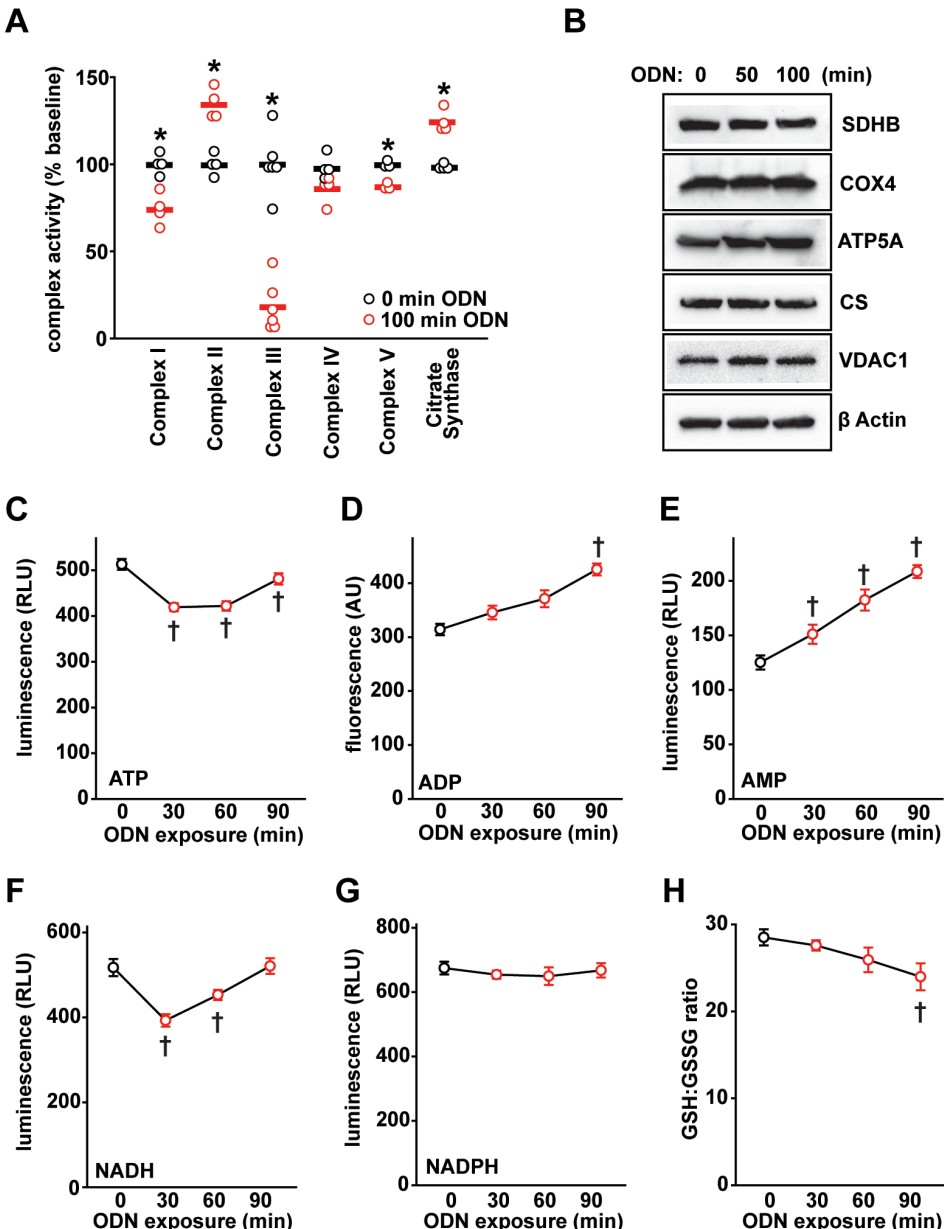

**Fig 2. Mitochondrial energy metabolism altered by CpG ODN. (A)** Summary electron transport chain complex enzyme activity with ODN treatment of HBEC3-KT cells. (**B**) Mitochondrial protein immunoblots from lysates of cells treated with ODN. Relative abundance ATP (**C**), ADP (**D**), and AMP (**E**) in whole cell lysates at the indicated time points after ODN treatment. Relative abundance of NADH (**F**) an NADPH (**G**) in whole cells after ODN treatment. (**H**) Ratio of reduced:oxidized glutathione in HBEC3-KT cells treated with ODN. * p<0.003 vs PBS; † p<0.001 vs PBS. SDHB, succinate dehydrogenase subunit B; COX4, cytochrome c oxidase subunit IV; ATP5A, ATP synthase subunit alpha; CS, citrate synthase; VDAC1, voltage dependent anion channel 1; GSH, reduced glutathione; GSSG, oxidized glutathione.

Seeking to identify a TLR9-independent mechanism by which ODN alters mitochondrial energy metabolism, we investigated whether ODNs could directly stimulate mtROS production in isolated mitochondria. Remarkably, we found that direct ODN treatment of mitochondria isolated from HBEC3-KT cells recapitulates the inducible mtROS generation and

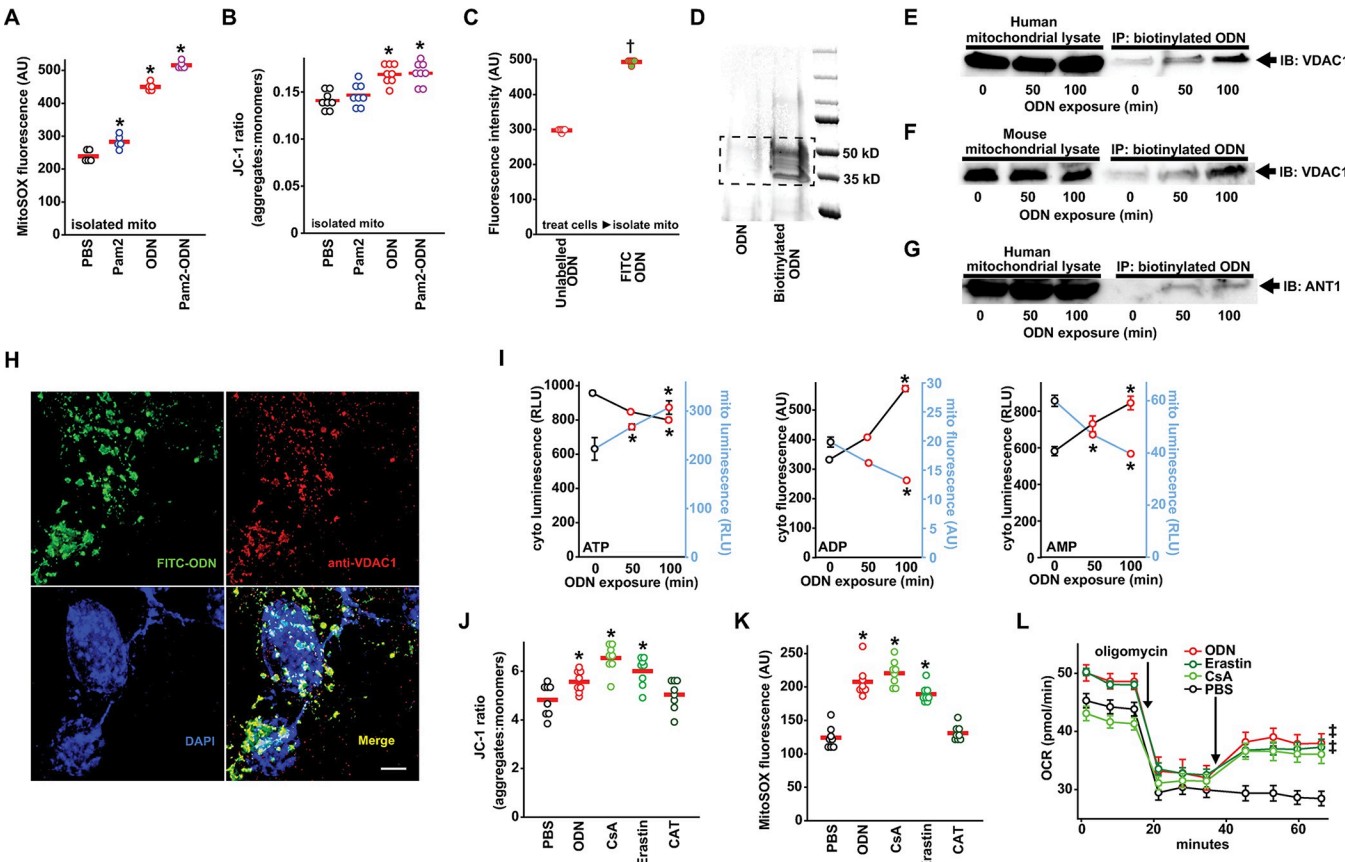

**Fig 3. Blocking mitochondrial nucleotide transition by CpG ODN leads to generation of antimicrobial mtROS.** (**A**) mtROS production and (**B**) $\Delta_{\Psi m}$ increase in isolated mitochondria that were treated with Pam2, ODN or both. (**C**) Fluorescence intensity of mitochondria isolated from HBEC3-KT cells treated with FITC-labeled or unlabeled ODN. (**D**) Mitochondria were isolated from the biotinylated ODN-treated HBEC3-KT cells, and streptavidin precipitants from mitochondrial lysates were resolved by polyacrylamide gel electrophoresis and silver stained. Dashed line indicates bands excised for mass spectrometry analysis. As in **D**, streptavidin precipitants were probed for VDAC1 in mitochondrial lysates from (**E**) human or (**F**) mouse cells following treatment with ODN for the indicated time. Mitochondrial lysates from human cells were precipitated and probed for ANT1 (**G**) following treatment with ODN for the indicated time. (**H**) Representative images of HBEC3-KT cells treated with FITC-labeled ODN then stained with Alexa Fluor 555-labeled anti-VDAC1 antibody. Scale bar, 100 μm. (**I**) Measurements of cytosolic and mitochondrial levels of ATP, ADP & AMP in ODN-treated HBEC3-KT cells at the indicated times. Mitochondrial membrane potential $\Delta_{\Psi m}$ (**J**) and mtROS (**K**) 100 min after HBEC3-KT treatment with the indicated mitochondrial permeability modulators. (**L**) Seahorse analysis of oxygen consumption following the indicated mitochondrial permeability modulators in oligomycin-inhibited HBEC3-KT cells, shown as mean ± SEM. * $p < 0.001$ vs PBS by ANOVA; † $p < 0.001$ vs unlabeled ODN by two-way Student's t test. ‡ $p < 0.0006$ vs PBS-treated by one-way ANOVA using Dunnett's test for multiple comparisons Mito, mitochondria; VDAC1, voltage dependent anion channel 1; ANT1, adenine nucleotide translocator 1; CsA, cyclosporin A; CAT, carboxyatractyloside; OCR, oxygen consumption rate.

increased $\Delta_{\Psi m}$ observed in whole cells (Fig 3A and 3B). Alternately, to examine whether ODN interacts with mitochondria in intact cells, whole cells were treated with fluorescently-labeled ODN, then the mitochondria were isolated and assessed for fluorescence intensity. As in Fig 3C, mitochondria from cells treated with labeled ODN displayed significant fluorescence, supporting a hypothesis that ODN can directly interact with mitochondria to stimulate mtROS production. To identify potential mitochondrial ODN binding partners, HBEC3-KT cells were treated with biotin-labeled ODN, then the streptavidin precipitants from mitochondria lysates were resolved on an SDS PAGE gel. The silver stain in Fig 3D demonstrates bands present only after ODN treatment. The boxed area was excised and liquid chromatography–mass spectrometry proteomic analysis generated a list of candidate targets (S1 Table).

Among the most differentially detected peptides in the bands cut from Fig 3D was voltage-dependent anion-sensitive channel 1 (VDAC1), an outer mitochondrial membrane protein

component of the VDAC1-ANT1-mCK complex that regulates exchange of ATP and ADP between the mitochondria and cytosol [25,26]. In targeted pulldown studies in mouse and human cells, we confirmed that VDAC1 was detected in immunoprecipitated samples of biotinylated ODN treated cells (Figs 3E–3F and S7). The association of ODN with VDAC1 increased with treatment time. We also detected association of ODN with adenine nucleotide translocator 1 (ANT1), the inner mitochondrial membrane component of the VDAC1-ANT1-mCK complex (Figs 3G and S7). When HBEC3-KT cells were treated with fluorescently labeled ODN then VDAC1 was localized with fluorescently labeled antibody (Fig 3H), 75% of pixels occupied by VDAC1 were also occupied by ODN. The mean Pearson's correlation coefficient between ODN and VDAC1 pixel intensity was 0.84 (S8 Fig), indicating a high degree of colocalization in intact cells.

Given the role of the VDAC1-ANT1-mCK complex as an ATP:ADP antiporter, we investigated whether altered ATP/ADP localization might account for the changes in whole-cell energy stores previously observed following ODN treatment (Fig 2C–2E). Indeed, we found that ODN treatment causes mitochondrial ATP levels to rapidly increase and cytosolic ATP levels to precipitously decline, with the opposite pattern for ADP and AMP (Fig 3I), consistent with an ODN-induced blockade of ATP:ADP antiporter function. To test whether VDAC1 antagonism can explain the ODN effect on mitochondrial energy metabolism, we investigated the effects of a known VDAC1 inhibitor erastin [27,28] and an ANT1 inhibitor carboxyatractyloside (CAT) [29, 30]. Because VDAC1 is one of the subunits composed of the mitochondrial permeability transition pore (mPTP), we also exposed cells to mPTP inhibitor cyclosporin A [31,32]. As shown in Fig 3J–3L, erastin and cyclosporin A caused changes in $\Delta\Psi_m$, mtROS production, and oxygen consumption that were comparable to ODN, suggesting that blocking VDAC-mediated mitochondrial nucleotide transition exerts these effects. Similar results were observed with an alternate VDAC1 inhibitor, VBIT-4 (S9 Fig). When using pharmacologic inhibitors, the possibility that off-target effects contribute to the observed results must be considered. However, given the similar responses observed across the tested agents, off-target effects seem unlikely to fully explain these cellular responses.

## AMPK-dependent metabolic changes increase electron delivery to complex II

AMP-activated kinase (AMPK) is a cellular energy sensor that is activated when cytosolic AMP levels rise due to ATP consumption [33,34], in turn activating catabolic pathways to promote ATP production, including acetyl-CoA carboxylase (ACC) [35,36]. Congruent with the observations that ODN alters cellular ATP/ADP/AMP stores (Figs 2 and 3), reverse phase protein array analysis of HBEC3-KT cells treated with ODN [37] revealed AMPK and ACC to be among the most activated signaling pathways following ODN exposure (Fig 4A). We confirmed time-dependent ODN-induced phosphorylation of AMPKα1, AMPKα2 and ACC in vitro (Figs 4B and S10). Similarly, AMPKα1 phosphorylation was induced by both erastin and cyclosporin A in vitro (Figs 4C and S10). ODN-induced AMPKα phosphorylation was also demonstrated by immunofluorescence staining in mouse airways and alveolar space (Fig 4D–4E). When AMPKα genes were conditionally deleted (S11 Fig), ODN-inducible mtROS production was significantly reduced without impacting baseline mtROS production (Fig 4F).

A principle means by which AMPK-ACC pathway activation promotes mitochondrial energy production is through increased carnitine palmitoyltransferase 1 (CPT1)-dependent fatty acid β-oxidation. We found that ODN treatment increases acetyl CoA concentrations and fatty acid β-oxidation (Fig 5B–5C). This ODN-induced fatty acid oxidation is abrogated when AMPK is depleted (Fig 5D). Treatment of HBEC3-KT cells with etomoxir, an irreversible inhibitor of CPT1 [38,39], significantly attenuated ODN-induced mtROS production,

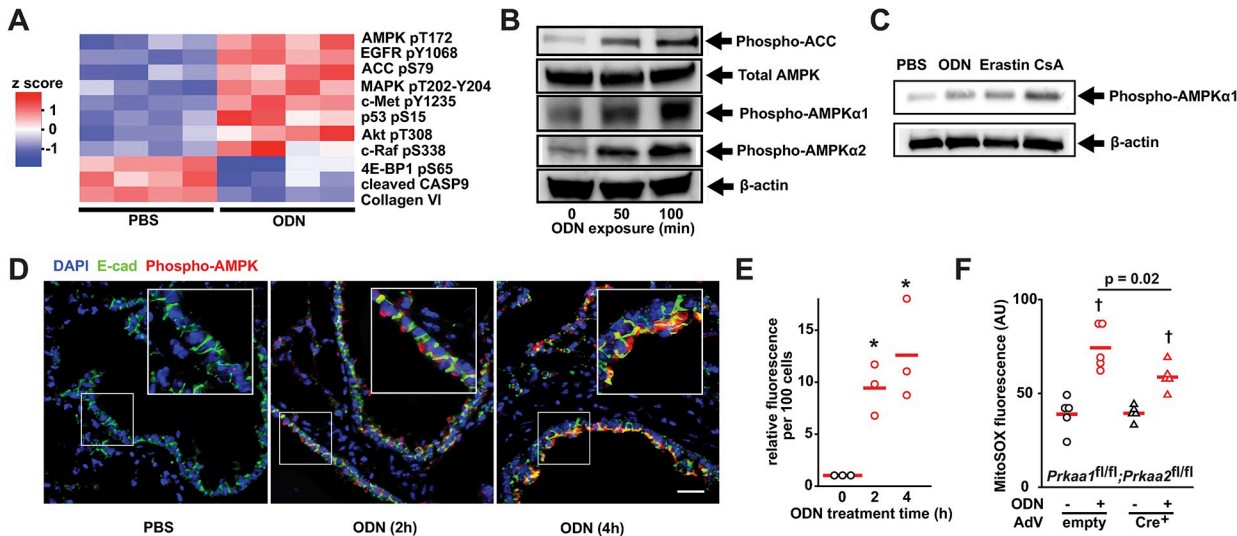

**Fig 4. ODN-induced AMPK activation.** (**A**) RPPA heatmap from HBEC3-KT cells treated with PBS or ODN. (**B**) Immunoblot of AMPK and ACC proteins after ODN treatment. (**C**) Immunoblot for phospho-AMPKα1 following treatment with the indicated mitochondrial permeability modulators in HBEC3-KT cells. (**D**) Phospho-AMPKα immunofluorescence in mouse lungs after treatment with ODN. Boxed area enlarged in insert. Scale bar, 100 mm. (**E**) Quantification of fluorescence in **D**. (**F**) mtROS production in primary *Prkaa1*$^{fl/fl}$;*Prkka2*$^{fl/fl}$ mouse tracheal epithelial cells infected with empty or Cre$^+$ adenovirus, then treated with PBS or ODN. * p <0.01 vs 0 min; † p <0.001 vs. (syngeneic) PBS treated. RPPA, reverse phase protein array; AMPK, AMP-activating protein kinase; ACC, acetyl-CoA carboxylase; AdV, adenovirus.

oxygen consumption and ETC complex II activity (Fig 5E–5G). Knockdown of *CPT1* (S12 Fig) also attenuated inducible mtROS production to a similar degree to etomoxir (Fig 5H).

Mitochondrial fatty acid β-oxidation generates NADH and FADH$_2$, which contribute electrons to the ETC via electron shuttle proteins. Specifically, FADH$_2$-carried electrons are transferred to coenzyme Q (CoQ) by electron flavoprotein dehydrogenase (ETFDH) [40], while NADH-carried electrons are transferred to complex II by glycerol-3-phosphate dehydrogenase (GPD2) [41]. Knocking down either of these shuttles (S12 Fig) attenuated ODN-induced mtROS production (Fig 5I–5J). Fatty acid β-oxidation also generates acetyl-CoA which transfers electrons to the ETC via the tricarboxylic acid (TCA) cycle. Treatments with TCA intermediate metabolites oxaloacetate and α-ketoglutarate or the analogue dimethyl malonate attenuated ODN-induced mtROS production (S13 Fig). Dimethyl malonate and oxaloacetate are ETC complex II inhibitors while α-ketoglutarate inhibits glutaminolysis [42–44].

In the ETC, CoQH$_2$ is generated when CoQ accepts electrons from FADH$_2$. The CoQH$_2$:CoQ ratio has been described as an indicator of ETC efficiency, with an increased ratio associated with increased mtROS production [45]. As in Fig 5K, ODN treatment caused an increase in CoQH$_2$:CoQ ratio. Although augmented β-oxidation is required for maximal ODN-induced mtROS production, inducible mtROS production can also be partially attenuated by inhibiting glycolysis and/or glutaminolysis (S14 Fig). Future work will explore the contributions of these pathways.

Together, these results indicated that ODN-induced and AMPK-regulated metabolic modulation enhances electron delivery to ETC, increases complex II activity and eventually drives mtROS induction. As schematically summarized in Fig 5A, ODN-impaired ATP/ADP antiport lowers cytosolic ATP levels, serially activating AMPK, ACC and CPT-1, thereby promoting fatty acid β-oxidation that drives additional electrons to ETC complex II via the TCA cycle.

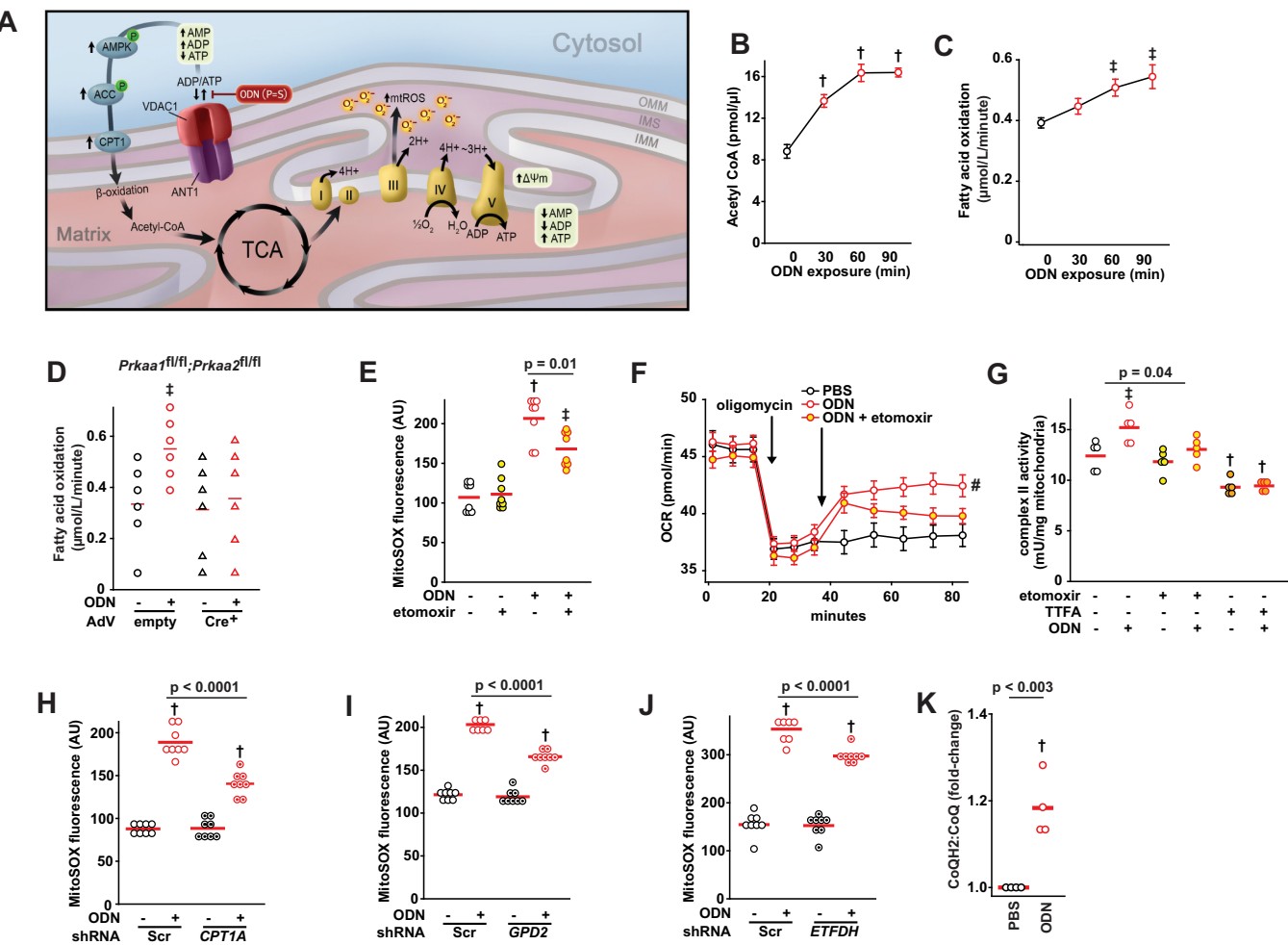

**Fig 5. ODN treatment increases electron delivery to complex II.** (**A**) Schematic overview of immunometabolic modulation by ODN. (**B**) Acetyl-CoA levels in ODN-treated HBEC3-KT cells. (**C**) Fatty acid oxidation after ODN treatment of different intervals. (**D**) ODN-induced fatty acid oxidation in AMPK-deficient cells. (**E**) mtROS production following treatment with ODN and/or β-oxidation inhibitor etomoxir. (**F**) Oxygen consumption following the indicated treatments, shown as mean p± SEM. (**G**) HBEC3-KT cell complex II activity following treatment with the indicated agents.ODN-induced mtROS production in cells with knockdowns of gene CPT1A (**H**) and the genes for electron shuttles GPD2 (**I**) or ETFDH (**J**). (**K**) Ratio of reduced:oxidized CoQ in mitochondria isolated from HBEC3-KT cells treated with PBS or ODN. † p <0.001 vs. (syngeneic) PBS treated; ‡ p < 0.05 vs (syngeneic) PBS treated. # p = 0.008 vs PBS-treated by one-way ANOVA using Dunnett's test for multiple comparisons. OCR, oxygen consumption rate; Scr, scrambled shRNA control; CPT1A, carnitine palmitoyltransferase 1A; GPD2, glycerol-3-phosphate dehydrogenase 2; ETFDH, electron transfer flavoprotein-ubiquinone dehydrogenase.

## ODN-induced mtROS generation at complex III is ΔΨm-dependent

It is intriguing that ODN simultaneously increases complex II activity and decreases complex III activity. We hypothesized that these changes in ETC complex activity might provide insights into the site(s) of mtROS generation. A series of ETC complex inhibitors were used to determine the roles of ETC complexes in ODN-induced mtROS generation.

mtROS are formed when molecular oxygen interacts with an electron leaked among electron transport chain [23], typically from complex I [flavin (F) site and ubiquinone reduction (Q) site] or complex III (Q$_o$ site) [46]. Whereas mtROS formation at complex I following other stimuli can be inhibited by rotenone (Q site) or diphenyleneiodonium (F site) [47, 48], neither agent impeded ODN-induced mtROS (S15 Fig). Similarly, while TCA cycle input can support reverse election transport from complex II to complex I in the setting of high $\Delta_{\Psi_m}$

[49], we found that neither succinate nor fumarate influence ODN-induced mtROS formation (S13 Fig). These findings, along with the modest ODN impact on complex I activity (Figs 2A and S3), indicated that complex I is not a major site of ODN-induced mtROS production. In contrast, while inhibitors of complex II activity reduced ODN-induced mtROS, inhibitors of complex III enhanced ODN-induced mtROS dramatically (S15 Fig). We thus concluded that complex III is the main ODN-induced mtROS generating site following forward electron transfer. Additionally, while complex IV inhibition had no impact on mtROS induction, the complex V inhibitor oligomycin decreased ODN-induced mtROS production. As oligomycin treatment collapses mitochondrial membrane potential $\Delta_{\Psi_m}$, this result suggested that ODN-induced $\Delta_{\Psi_m}$ increases may be required for mtROS formation.

We compared the impacts of ODN and complex III inhibitor antimycin on complex III activity and mtROS induction. Although treatment with antimycin or ODN resulted in similar inhibition of complex III enzymatic activity and initial mtROS production (Fig 6B and 6C), the two agents function differently. When stigmatellin and myxothiazol inhibit electron transfer from complex III to complex IV by binding the $CoQH_2$ (ubiquinol) oxidation ($Q_o$) site, antimycin cannot induce further oxygen consumption for superoxide production while ODN still can (Fig 6D). Antimycin can induce rapid induction of mtROS production in intact epithelial cells and isolated mitochondria, but this effect plateaus by 40 min. Conversely, ODN-induced mtROS continued to increase throughout the period of exposure in intact cells and isolated mitochondria, suggesting different mechanisms of mtROS generation (Fig 6E and 6F). Central to these differences appeared to be their opposing effects on $\Delta_{\Psi_m}$. In both whole cell and isolated mitochondria models, ODN induced increased $\Delta_{\Psi_m}$ while antimycin reduced $\Delta_{\Psi_m}$ below that observed in sham-treated samples (Fig 6G and 6H). Disrupting $\Delta_{\Psi_m}$ with the uncoupler FCCP (Fig 6I–6J) significantly impaired ODN-induced mtROS generation but had little effect on antimycin-induced mtROS generation (Fig 6K and 6L). FCCP treatment demonstrated that ODN-induced oxidation of cytochrome $b_H$ is $\Delta_{\Psi_m}$-dependent, whereas FCCP did not alter the oxidation of cytochrome $b_H$ in antimycin-treated mitochondria (Fig 6M). Congruently, in isolated mitochondria, FCCP reversed ODN-impaired complex III electron transfer activity but had no such effect on antimycin treated mitochondria (Fig 6N). The generation of ODN-induced mtROS at complex III is graphically displayed in Fig 6A. Under homeostatic conditions, complex III quickly transfers $CoQH_2$-carried electrons to cytochrome c1, which, in turn, transfers the electrons to complex IV. This process facilitates proton pumping across the inner mitochondrial membrane and establishes normal $\Delta_{\Psi_m}$ [50,51]. However, ODN-induced increase in $\Delta_{\Psi_m}$ hinders the proton pumping, impeding electron transfer at the $Q_o$ and quinone reduction ($Q_i$) sites. This $\Delta_{\Psi_m}$-dependent retardation of electron transfer in complex III, in coordination with an increased forward electron transfer from complex II, increases the likelihood that highly-reactive electrons will "leak" to interact with free oxygen, resulting in increased formation of mtROS, in the form of superoxide, at complex III [52–54].

Thus, while dissecting the process of ODN-induced antimicrobial mtROS formation, we identified that mtROS induction requires both AMPK-directed metabolic reprograming to augment electron delivery to ETC complex II (Fig 5) and increased $\Delta_{\Psi_m}$ to retard electron transfer at complex III (Fig 6).

## mtROS induction yields TLR9-independent antimicrobial effects

To demonstrate the protective effects of antimicrobial mtROS induced by ODN, we have shown that scavenging mtROS by mitoTEMPO or mitoQ (S16A and S16B Fig) significantly decreases the bacterial killing induced by Pam2-ODN combined treatment in HBEC3-KT cells [18]. In this system, ODN-induced mtROS levels were not appreciably influenced by the

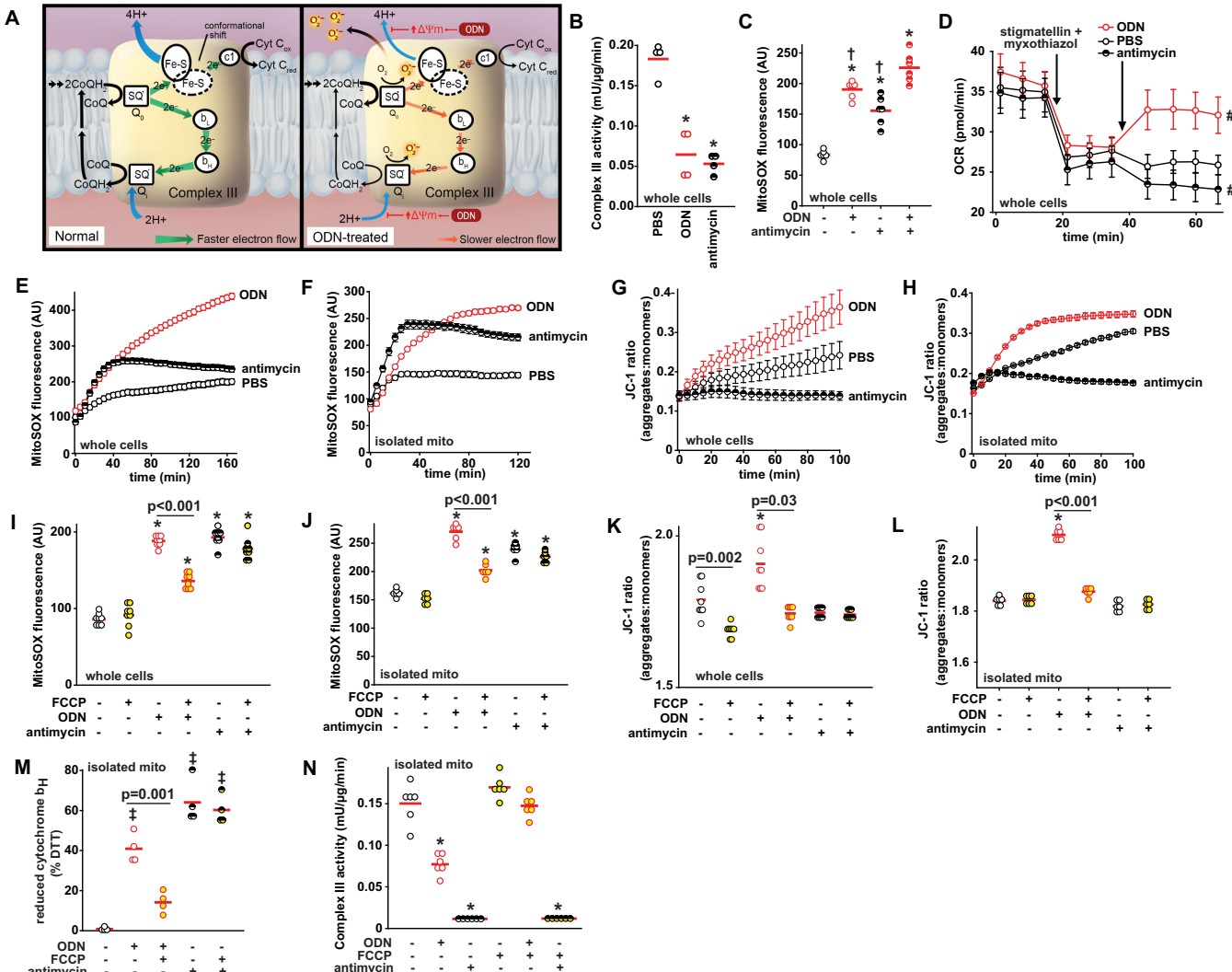

**Fig 6. mtROS formation at complex III is ΔΨm-dependent.** (**A**) Schematic model of $\Delta_{\Psi m}$-dependent mtROS formation at complex III. (**B**) Electron transport chain complex III activity in HBEC3-KT cells 100 min after ODN or antimycin treatment. (**C**) mtROS production 100 min after the indicated treatments. (**D**) Oxygen consumption following the indicated treatments in stigmatellin and myxothiazol-inhibited HBEC3-KT cells, shown as mean ± SEM. Time course of mtROS generation following PBS, ODN or antimycin treatment in (**E**) HBEC3-KT cells or (**F**) isolated mitochondria. Time-dependent mitochondrial membrane potential in (**G**) HBEC3-KT cells or (**H**) isolated mitochondria. Mitochondrial membrane potential $\Delta_{\Psi m}$ 100 min after ODN or antimycin treatments in (**I**) HBEC3-KT cells or (**J**) isolated mitochondria with or without FCCP pre-treatment. mtROS generation 100 min after ODN or antimycin treatments in (**K**) HBEC3-KT cells or (**L**) isolated mitochondria with or without FCCP pre-treatment. (**M**) Reduced mitochondrial complex III cytochrome $b_H$ levels following ODN or antimycin treatment with or without FCCP pre-treatment, expressed relative to DTT-treated mitochondria (DTT-treated presumed 100% reduced). (**N**) Complex III activity in isolated mitochondria 15 min after the indicated treatments. * p<0.001 vs PBS by ANOVA; † p<0.008 vs ODN + antimycin treated by ANOVA; ‡ p<0.02 vs PBS by ANOVA. # p < 0.009 vs PBS-treated by one-way ANOVA using Dunnett's test for multiple comparisons. mito, mitochondria; DTT, dithiothreitol.

presence of bacteria nor were broader measures of cellular ROS (S16C and S16D Fig). While pre-treatment with either an ETC complex II inhibitor TTFA or the $\Delta_{\Psi m}$ uncoupler FCCP alone inhibited ODN-induced mtROS to some extent, TTFA-FCCP combination treatment maximally inhibited mtROS production in HBEC3-KT cells (S17 Fig) and reversed the mitochondrial reduced:oxidized ratio of ODN-treated cells (Fig 7A and 7B). As shown in Fig 7C and S17 Fig, the bacterial killing induced by Pam2 and ODN was obviated in HBEC3-KT cells when the cells were pretreated with TTFA-FCCP. Congruently, wild type mice with impaired

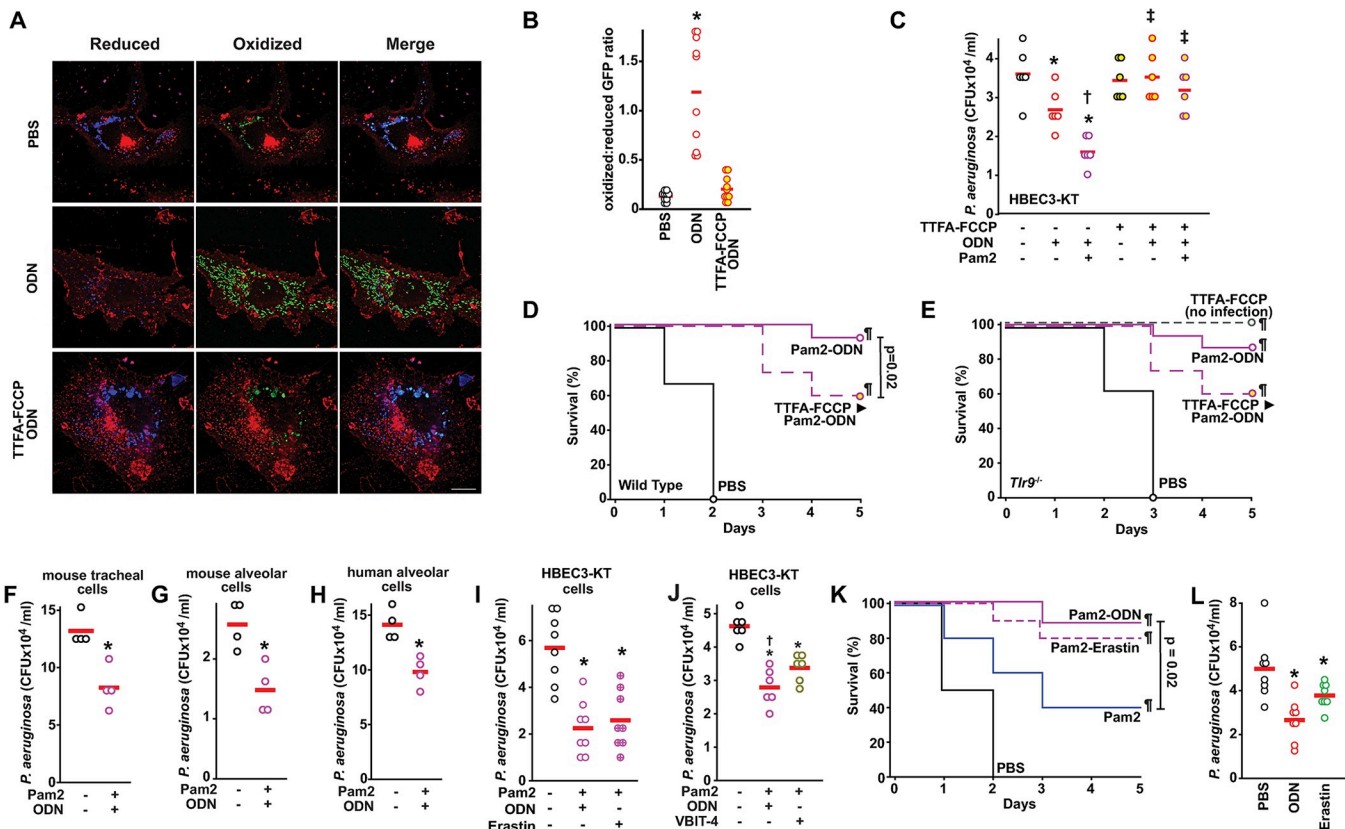

**Fig 7. mtROS induction stimulates antimicrobial responses.** (**A**) Representative fluorescence images primary tracheal epithelial cells harvested from mt-roGFP mice, pre-treated (or not) with TTFA and FCCP, then treated with PBS or ODN. Images shown as gradient of color intensity from the reduced (blue) form to the oxidized (green) form of roGFP. Scale bar, 50 μm. (**B**) Ratio of the fluorescence intensity of the oxidized:reduced roGFP from **A**, quantified at 488 nm and 405 nm, respectively. (**C**) Bacterial burden of HBEC3-KT cells treated with the indicated ligands with or without TTFA-FCCP treatment. (**D**) Survival of wild type mice challenged with *P. aeruginosa* one day after nebulized treatment with PBS or Pam2 and ODN with or without TTFA-FCCP (n = 15 mice/group). (**E**) Survival of *Tlr9*[-/-] mice challenged with *P. aeruginosa* one day after nebulized treatment with PBS or Pam2 and ODN with or without TTFA-FCCP (n = 15 mice/group). Bacterial burden of primary mouse tracheal epithelial cells (**F**), primary mouse alveolar cells (**G**), or primary human alveolar cells (**H**) after treatment with Pam2 and ODN. Bacterial burden of HBEC3-KT cells treated with Pam2 and erastin or ODN (**I**) or Pam2 and VBIT-4 or ODN (**J**). (**K**) Mouse survival of *P. aeruginosa* challenge given one day after nebulized treatment with the indicated agents (n = 15 mice/group). (**L**) Mouse lung bacterial burden immediately after *P. aeruginosa* challenge following treatment with the indicated agents (n = 4 mice/group). * p <0.02 vs PBS, † p< 0.05 vs ODN, ‡ p < 0.05 vs same ligand without TTFA-FCCP, ¶ P <0.0001 vs. PBS.

lung epithelial mtROS generation due to aerosolized TTFA-FCCP pretreatment prior to treatment with Pam2 and ODN were less protected against *P. aeruginosa* pneumonia than were mice who received sham aerosol pretreatment prior to receiving Pam2-ODN (Fig 7D). Notably, both the Pam2-ODN-induced *P. aeruginosa* pneumonia protection and the TTFA-FCCP-induced impairment were observed in *Tlr9* knockout mice (Fig 7E). Consistent with our prior reports [17], airway delivery of TTFA-FCCP had no observed systemic effects on the mice in the absence of infection (Fig 7E). Congruent with their ability to generate ODN-induced mtROS (S2 Fig), primary lung epithelial cells from mouse trachea, mouse alveoli and human alveoli treated with Pam2 and ODN exert bacterial killing effects (Fig 7F–7H). In light of our data supporting VDAC1 blockade as central mediator of protection, we tested whether erastin or VBIT-4 substituted for ODN could protect against infection. In combination with Pam2, erastin and VBIT-4 induced comparable bacterial killing to ODN by HEBC3-KT cells (Fig 7I–7J). Strikingly, when delivered by aerosol with Pam2, erastin yielded a similar survival advantage to ODN following *P. aeruginosa* pneumonia challenge in mice (Fig 7K). Even when

delivered without Pam2, both ODN and erastin induced significant reductions in the lung bacterial burden (Fig 7L).

In summary, we found that antimicrobial mtROS are generated from lung epithelial cells following ODN treatment. ODN interacts with VDAC1 to alter mitochondrial nucleotide transport, driving AMPK-ACC-CPT1-mediated electron delivery to ETC complex II and increasing $\Delta\Psi_m$ to promote superoxide production at complex III.

## Discussion

Synthetic CpG ODNs have been explicitly developed as immunomodulators and adjuvants [24, 55, 56]. ODN CpG motifs mimic naturally-occurring pathogen-associated molecular patterns recognized by TLR9 that initiate NF-κB-dependent antimicrobial signaling cascades [57–59]. Here, we report previously unknown, TLR9-independent immunometabolic modulation by ODN that results in generation of pneumonia-protective mtROS.

ODN-induced mtROS formation is fundamentally a manifestation of altered energy metabolism. ODN interacts with VDAC1 and ANT1 localized in the mitochondrial membranes. The VDAC1-ANT1-mCK complex regulates the exchange of metabolites between the mitochondria and cytosol [25,26]. We show that VDAC1 binding ODN perturbs cellular nucleotide distribution, activating the AMPK-ACC pathway, and promoting fatty acid β-oxidation. Fatty acid β-oxidation augments $FADH_2$-carried electron flux to the ETC by ETFDH, GPD2, and the TCA cycle, all converging on the CoQ pool ($CoQH_2$ and CoQ). Disruption of any of these elements leads to decreases in inducible mtROS generation.

The increased electron delivery as a consequence of AMPK-ACC activation results in increased complex II activity, however, these observations do not resolve why complex III activity are decreased. During normal bifurcated electron transfer in complex III [60,61], semiquinone ($SQ\bullet^-$) intermediates forming at the Qo or Qi site instantaneously transfer electrons to the low potential heme b (bL) or CoQ, minimizing electron leakage. However, under certain conditions, accumulation of $SQ\bullet^-$ increases electron leak [52]. In one example, antimycin inhibits CoQ reduction at the $Q_i$ site, leading to electron accumulation on cytochrome b hemes, allowing $SQ\bullet^-$ more time to interact with molecular oxygen to form superoxide at the $Q_o$ site [62,63]. Alternatively, high $\Delta\Psi_m$ attenuates the proton pump, retarding electron transfer and sustaining cytochrome b hemes in reduced states that cause accumulation of $SQ\bullet^-$ at the $Q_o$ and $Q_i$ sites [53,64]. Here, our findings support the latter as the responsible mechanism as uncoupling $\Delta\Psi_m$ with FCCP reduces ODN-induced mtROS and reverses ODN-impaired complex III activity. Thus, $\Delta\Psi_m$ accentuation by VDAC-perturbed ATP accumulation in mitochondria increases mtROS production and impairs complex III activity.

Although ROS production is often regarded as an untoward cellular event that contributes to degenerative diseases [65,66], there is robust evidence that controlled mtROS generation contributes to critical signaling events in a wide range of physiologic processes that extend host survival [67–71], including by augmentation of protective antimicrobial responses [72–74]. Superoxide formed at the complex III $Q_o$ site may be particularly well suited to function as a cytosolic signaling molecule, as the $Q_o$ site is adjacent to the intermembrane space with about half of its superoxide diffusing to the cytoplasmic side of the inner membrane [75,76]. Here, we demonstrate that complex III-dependent mtROS induction is required for maximally ODN-induced bacterial killing in vitro and in vivo. Although the current work does not explicitly test whether mtROS directly kill pathogens or act as cell signals to initiate antimicrobial responses, both topics are areas of active investigation. We have previously shown that ROS induction is associated with improved host and epithelial survival of infections and is well-tolerated by epithelial cells in the absence of an infectious challenge [6,17]. Although the current

data focus on the mechanisms of protective mtROS induction in lung epithelial cells, induction of dual oxidase-dependent ROS production is also required for maximal protection and non-epithelial cells can also contribute to the protective response to therapy, though we have shown that epithelial cells are essential to the protection [6,8].

In summary, we identify metabolic mechanisms underlying the ODN-induced antimicrobial mtROS formation. Under physiologic conditions, mtROS production is exquisitely tightly regulated, but we show here that therapeutic manipulation of mtROS is achievable, protective against otherwise lethal infections, and well tolerated by the host. Indeed, this intervention has also been safely tested in five completed human trials (NCT04313023, NCT04312997, NCT03794557, NCT02566252, NCT02124278) with more in preparation. Because of our interest in pneumonia, all of the current work is performed in lung epithelial cells, but we anticipate similar responses can be detected in other epithelial cells and, likely, other cell types.

## Methods

### Primary cell cultures and cell lines

To isolate mouse tracheal epithelial cells (mTECs), mice were anesthetized and tracheas were excised and digested in 1.5 mg/ml Pronase overnight at 4˚C. mTECs were harvested by centrifugation and then cultured on collagen coated tissue culture plates or transwells in Ham's F12 media supplemented with differentiation growth factors and hormones as previously described [77].

Mouse alveolar epithelial cells were harvested as previously described [78,79]. Briefly, after flushing the right ventricle with PBS, the lungs were minced with forceps, then digested in Leibovitz's L-15 medium with 2 mg/mL collagenase type I, 2 mg/mL elastase, and 0.5 mg/mL DNase I for 30 min at 37˚C. To stop digestion, 20% fetal bovine serum (FBS) was added and the tissue was mechanically homogenized, then placed on ice in a cold room. The tissue was filtered with a 70 μm cell strainer, and centrifuged at 5000 rpm for 1 min, resuspended in red blood cell lysis buffer for 3 min, washed with Leibovitz's media with 10% FBS and filtered into a 5 ml glass tube with a cell strainer cap. Sample were incubated for 30 minutes with CD45-PE/Cy7, ICAM2-A647, E-Cadherin-A488 at a concentration of 1:250. Then, the cells were centrifuged at 5000 rpm for 1 minute and washed in Leibovitz's with 10% FBS. The cells were resuspended in this same solution and filtered through a cell strainer cap. Before sorting on an Aria II Cell Sorter, SYTOX Blue was added. For gating, single cells were selected and dead cells excluded. CD45 and ICAM2 negative cells were selected, and from that population ECAD positive cells were selected. Alveolar epithelial cells were seeded on poly-L-Lysine coated transwells at 20,000 cells/cm$^2$ and grown at air-liquid interface culture with alveolar epithelial cell medium for 5 days.

Normal human bronchial epithelial (NHBE) cells were purchased from American Type Culture Collection (ATCC, Manassas, VA) and cultured in airway epithelial cell basal medium supplemented with bronchial epithelial cell growth kit (ATCC, Manassas, VA). Normal human Type II alveolar epithelial Cells were purchased from AcceGen (Fairfield, New Jersey) and air-liquid interface cultured in poly-L-Lysine coated transwells with alveolar epithelial cell medium (ScienCell Research Laboratories, Carlsbad, CA) for 5 days. Cells were seeded at a density of 20,000 cells/cm$^2$ on transwells.

Immortalized Human bronchial epithelial (HBEC3-KT) cells were kindly provided by Dr. John Minna. Murine lung epithelial (MLE-15) cells were kindly provided by Dr. Jeffrey Whitsett. HBEC3-KT and MLE-15 cells were authenticated by the UT MD Anderson Characterized Cell Line Core Facility and IDEXX Bioresearch (Columbia, MO), respectively. HBEC3-KT cells were cultured in keratinocyte serum-free medium supplemented with human epidermal

growth factor and bovine pituitary extract (Thermo Fisher Scientific, Grand Island, NY). MLE-15 cells were cultured in DMEM/F2 medium supplemented with 2% of fetal bovine serum and 0.5% of Insulin-Transferrin-Selenium (Thermo Fisher Scientific, Grand Island, NY). Cell cultures were maintained in the presence of 1% of penicillin/streptomycin and gluta-mine. All cells were cultured at 37˚C with 5% $CO_2$. All human cell experiments were per-formed in accordance with Institutional Review Board of The University of Texas MD Anderson Cancer Center (MDACC).

## Mice

Wild type C57BL/6J mice were purchased from The Jackson Laboratory (Bar Harbor, ME). *Prkaa1fl* and *Prkaa2fl* mice were purchased from Jackson. *TLR9−/−* mice were provided by Dr. Shizuo Akira [80]. *CMV mt-roGFP* mice were generated by Dr. D James Surmeier and kindly provided by Dr. Farhad Danesh [19,20]. *Sftpc-Cre* mice were kindly provided by Dr. Brigid Hogan [81]. All mouse experiments were performed in accordance with the MDACC Institu-tional Animal Care and Use Committee.

## In vivo infection model

As previously described [8,18], 10 ml of PBS, single ligand 4 μM Pam2CSK4, single ligand 1 μM ODN M362, or a combination of both ligands in 1× PBS was placed in an Aerotech II nebulizer (Biodex, Shirley, NY) and delivered to unrestrained mice in an exposure chamber via an influx polyethylene tube over 40 min. Nebulization was driven by 10 L/min air supple-mented with 5% $CO_2$. The exposure chamber connects with an identical efflux polyethylene tube with a low resistance microbial filter (BB50T, Pall, East Hills, NY) at its end vented to a biosafety hood.

*Pseudomonas aeruginosa* strain PA103 was purchased from ATCC and stored as frozen stock ($1×10^8$ colony-forming unit CFU/ml) in 20% glycerol in Luria-Bertani (LB) medium. Typically, 1 ml of frozen stock was incubated overnight in 100 ml of Tryptic Soy Broth (TSB) at 37˚C with 5% $CO_2$, then expanded in 1 liter of fresh LB media at 37˚C to OD 600 of 0.52. Bacterial suspensions were centrifuged, washed, re-suspended in 1× PBS, and aerosolized using the same nebulization system for Pam2-ODN treatment over 20 min. For all bacterial challenges, a nebulized inoculum of 10 ml of ~$2×10^{10}$ CFU/ml were delivered. If not specified, 6 to 8 weeks old single sex mice were used for in vivo infection conducted in a BSL2 biohazard lab. Immediately following bacterial challenge, some mice were anesthetized and mouse lungs were harvested and homogenized using a Mini-Beadbeater-1 (Biospec, Bartlesville, OK). The lung homogenates were used to count lung colony-forming units (CFUs). The remaining mice were closely monitored for 12 days. The relevant euthanasia-triggering criteria consist of any evidence of distressed behaviors including hypothermia, impaired mobility, respiratory dis-tress, and inability to access food or water. When mice were identified to meet the criteria, they were subjected to euthanasia immediately. At least 8–10 mice per condition were evalu-ated for survival analysis, 4 mice per condition were sacrificed for pathogen burden assess-ment. Challenges were performed a minimum of 3 times.

## In vitro pathogen killing assay

Human or mouse primary lung epithelial cells at air-liquid interface on transwells or HBEC3-KT cells or MLE-15 cells on 6-well plates in complete media were cultured until cell growth reached ~80% confluence. Cells were replaced with fresh, antibiotic-free media con-taining PBS, Pam2, ODN or Pam2-ODN. The final concentrations of Pam2 or ODN in media were 2.4 μM or 0.6 μM, respectively. 4 h after the treatment, 20 μl of *P. aeruginosa* PA103

($1 \times 10^5$ CFUs/ml) were added to each culture well. 4 h after bacteria inoculation, 20 µl of supernatant from each well was aspirated, serially diluted, plated on a TSB agar plate and incubated for 16 h at 37°C. Bacterial CFUs were counted after the incubation [18]. Studies were performed a minimum of 3 times with 4 biological replicates per condition.

## Mitochondrial ROS detection, scavenging and inhibition

To detect mtROS generation, cells were incubated with 5 µM of each indicated detector, MitoSOX red, ROSstar 550, MitoNeoD, MitoTracker Red CMXRos or tetramethylrhodamine (TMRM) in a black-walled, clear bottomed 96-well plate for 1/2 h before ODN or PBS treatment [17]. A 24-well plate was used in air liquid interface culture with transwells. After fluorescent mtROS detectors were washed off, fluorescence was continuously measured on a BioTek Synergy2 plate reader for 3 h immediately after ODN or PBS addition. Excitation/emission wavelengths for mtROS-detecting agents are 510 nm/580 nm. Studies were performed a minimum of 3 times with 8 biological replicates per condition.

To scavenge mtROS, HBEC3-KT cells were exposed to 100 nM MitoTEMPO or 10 µM MitoQ for 1 h prior to fluorescent mtROS detector incubation and ODN or PBS treatment.

To disrupt mtROS production, HBEC3-KT cells were exposed to compounds that inhibit mitochondria electron transport chain activity. These include rotenone (10 µM), TTFA (200 µM), atpenin (10 nM), antimycin (100 nM & 5 µM), sodium azide (1 µM), oligomycin (2 µM) and FCCP (400 nM) etc., for 1 h prior to fluorescent mtROS detector incubation and ODN or PBS treatment. Inhibition of ODN-induced mtROS generation in vitro was achieved by concurrent application of TTFA (200 µM) and FCCP (200 nM).

To inhibit mtROS generation in vivo, mice were exposed to TTFA (200 mM) and FCCP (800 µM) in 10 ml of 50% DMSO solution in 1× PBS by nebulization. The 50% DMSO solution was nebulized as vehicle control. Mice received TTFA-FCCP or vehicle control were treated with either Pam2-ODN or PBS and then subjected to *P. aeruginosa* PA103 challenge.

## JC-1 assay

To monitor changes in mitochondria membrane potential ($\Delta\psi_m$) upon ODN treatment, HBEC3-KT cells were incubated with 2 µM of JC-1 fluorescence dye for 30 min. After the JC-1 dye was washed off, fluorescence was continuously measured with ODN or PBS treatment using a BioTek Synergy2 plate reader at wavelengths of 510nm/580nm for J-aggregates and 490nm/525nm for J-monomers. A higher ratio of J-aggregates to J-monomers indicates a higher $\Delta\psi_M$. To inhibit mitochondrial membrane polarization, TTFA (200 µM) and FCCP (200 nM) were concurrently applied for 1 h before JC-1 incubation and ODN or PBS treatment. Studies were performed a minimum of 3 times with 8 biological replicates per condition.

## Indirect immunofluorescence assay and co-localization image analysis

PBS or ODN-treated HBEC3-KT cells or MLE-15 cells growing on a chambered coverglass or frozen lung sections from PBS or ODN-treated mice were fixed with 2% paraformaldehyde, permeabilized with 0.1% Triton X-100, and blocked with 2% goat serum in 1× PBS. Cells or lung sections were incubated with primary antibodies against Phospho-AMPKα-1,2 or VDAC1 at a dilution of 1:200 for 1 h, then with AlexaFluor secondary antibodies (Life Technologies, Carlsbad, CA) at a dilution of 1:500 for half an hour, and counterstained with 4′,6-diamidino-2-phenylindole (DAPI) for 15 minutes. Cells were visualized using a DeltaVision deconvolution fluorescence microscope (GE Life Sciences). Fluorescence intensity of microscope images was quantified using ImageJ.

Pixel intensity data of VDAC1 and ODN-FITC in the fluorescence images were imported into MATLAB and all analysis was performed using default settings. Confidence intervals were set at the 95% confidence limit. Pixel intensity values for VDAC1 and ODN-FITC were compared directly on a scatterplot. Six independent analyses were performed. A simple linear regression model using the least squares standard approach was fit to the data. Pearson's correlation coefficients were calculated to determine whether VDAC1 pixel intensity tended to accumulate with ODN-FITC pixel intensity.

## Live cell fluorescence imaging

Fluorescent measurement of intracellular mtROS generation was carried out on live and metabolically active mTECs generated from *CMV mt-roGFP* mice. Cells growing on collagen coated chambered coverglass were mounted onto microscopic chamber at 37°C in air with 5% $CO_2$, treated with ODN or PBS and washed with 1× live cell imaging buffer (Life Technologies). CellMask DeepRed was added to stain the cell membrane. Images were obtained using a DeltaVision deconvolution fluorescence microscope (GE Life Sciences) at excitation wavelengths of 405nm and 488nm at 100 minutes post ODN treatment. Fluorescence intensity of microscope images was quantified using ImageJ software.

## Measurement of oxygen consumption rate

Oxygen consumption rate (OCR) was measured using a Seahorse XFe96 extracellular flux analyzer (Agilent, Santa Clara, CA). HBEC3-KT cells ($1.5 \times 10^4$ per well) were seeded into a XFe96 microplate and grew overnight in complete media. 1 h before the assay, the media was changed to Seahorse XF assay media (Agilent) and incubated in a non-$CO_2$ incubator at 37°C. The microplate was loaded onto the analyzer and basal respiration in these cells were recorded by real-time measurement of OCR. Then 25 μl of PBS, ODN and/or mitochondria inhibitors prepared in the assay medium were sequentially injected into each culture well (in 150 μl of assay media) via drug delivery ports. The final working concentrations of these testing regents in culture wells are ODN 1.2 μM, oligomycin 10 nM, and antimycin 1 μM. After injection of each regent, OCR was again measured. At the end of the assay, the number of viable cells was determined using trypan blue. OCR measurements were normalized to final cell numbers. OCR is expressed in pmole min$^{-1}$. Studies were performed a minimum of 3 times with 12 biological replicates per condition.

## Western blotting and immunoprecipitation

HBEC3-KT cells or MLE-15 cells were suspended in NP-40 lysis buffer containing Halt protease and phosphatase inhibitor cocktail (Millipore), disrupted by sonication, and extracted at 4°C for 30 min. The protein concentration of the lysate was determined using bicinchoninic acid (BCA) protein assay (Pierce). 50 μg protein in 1× Laemmli buffer was separated by SDS-PAGE and then transferred onto polyvinylidene difluoride (PVDF) membranes (Millipore). The PVDF membranes were blotted with primary antibodies, detected by secondary antibodies with conjugated horseradish peroxidase, and developed using a Pico-sensitive chemiluminescence kit (Pierce). All membranes were stripped and re-blotted for β-actin or GAPDH as loading control.

Whole cell or mitochondria lysates were prepared with biotinylated ODN-treated HBEC3-KT cells or MLE-15 cells. To precipitate proteins Bound by biotinylated ODN in vivo, streptavidin beads (Pierce) were incubated with whole cell or mitochondria lysates containing 300 μg protein overnight at 4°C under constant gentle rotating. After incubation, streptavidin beads were centrifuged, washed with 1× PBS containing 0.05% Tween-20, resuspended in

50 µl of 2× SDS loading buffer, and boiled for 10 minutes. Elutes from the streptavidin beads were loaded onto SDS PAGE gel (Bio-Rad) and immunoblotted with VDAC1 or ANT1 antibody.

## Lentiviral shRNA knockdown

GIPZ *E. coli* clones containing human *CPT1A*, *GPD2* and *ETFDH* lentiviral shRNA vectors were purchased from GE Dharmarcon (Lafayette, CO). The lentiviral shRNA vectors were purified using a QIAGEN plasmid kit. Lentiviruses bearing human *CPT1A*, *GPD2* or *ETFDH* shRNA were produced by co-transfection of the lentiviral shRNA vectors and lentiviral packaging vectors in 293T cells. The shRNA lentiviruses were collected and added into HBEC3-KT cell culture. Lentivirus-infected HBEC3-KT cells were selected by cell sorting based on GFP expression 3 days after infection. Efficiency of the shRNA knockdown was determined by immunoblot using anti-human *CPT1A*, *GPD2* or *ETFDH* antibodies.

## Adenoviral Cre knockout

Adenovirus containing Cre recombinase or control empty vectors were purchased from Viral Vector Core at the University of Iowa. As previously described [82], adenoviral infections were carried out in mTECs derived from *Prkaa1*$^{fl/fl}$;*Prkaa2*$^{fl/fl}$ mice.

## Mitochondria isolation

Mitochondria were isolated from either excised mouse lungs or harvested HBEC3-KT or MLE-15 cells. As previously described [83], a Polytron homogenizer (Pro Scientific, Oxford, CT) was used to dissociate tissues and cells. Mitochondria were separated from cytosol via serial centrifugation at 4˚C. The concentration of the isolated mitochondria was normalized in between PBS or ODN treatment groups using BCA protein assay (Pierce). The isolated mitochondria were maintained in mitochondrial isolation buffer on ice for further analysis.

## Biochemical analysis of mitochondrial complex activity

As previously described [84], spectrophotometric analysis of the respiratory chain complexes was performed on HBEC3-KT cells collected at the indicated time points after ODN treatment. The electron transport chain enzymes were assayed at 30˚ C using a temperature-controlled spectrophotometer (Ultraspec 6300 pro, Biochrom Ltd., Cambridge, England). The activities of complex I (NADH:ferricyanide reductase), complex II (succinate dehydrogenase), rotenone sensitive complex I+III (NADH:cytochrome c reductase), complex II + III (succinate:cytochrome c reductase) and complex IV (cytochrome c oxidase) were measured using appropriate electron acceptors/donors [85, 86]. The increase or decrease in absorbance of cytochrome c at 550 nm was measured for complex I + III, II + III, or complex IV. The activity of NADH:ferricyanide reductase was measured by oxidation of NADH at 340 nm. For succinate dehydrogenase, the reduction of 2,6-dichloroindophenol (DCIP) was measured at 600 nm. For citrate synthase, the reduction of dithionitrobenzoic acid (DTNB) was measured at 412 nm. Enzyme activities are expressed in nmol/min/mg protein. Complex III activity was measured using a Mitochondrial Complex III activity assay kit (BioVision). Complex V activity was measured using a Quantichrom ATPase assay kit (Bioassay Systems).

## Luminescence glo assay

HBEC3-KT cells (1×10$^4$ per well) were plated in an opaque-walled 96-well plate and grown overnight. Cells were treated with 0.6 µM ODN and collected at various time points after

treatment. Cells were lysed on the plate and incubated with luminescence glo regents per the luminescence glo assay kit's instructions (Promega). Luminescence was recorded using a Bio-Tek Synergy2 plate reader.

### Fatty acid oxidation assay

A non-radioactive fatty acid oxidation (FAO) assay kit (Biomedical Research Service, State University of New York at Buffalo) was used to measure FAO activity in ODN-treated HBEC3-KT cells. Cells were lysed and incubated with substrate octanoyl-CoA and then FAO assay solution in a 96-well plate, following the manufacturer's instruction. Oxidation of the octanoyl-CoA by the cell lysate generates NADH, which is coupled to the reduction of the tetrazolium salt INT to formazan in the FAO assay solution. The absorbance of the newly formed formazan (at 490 nm) is proportional to FAO activity.

### Coenzyme $Q_{10}$ measurement by triple quadruple LC-MS

Levels of ubiquinone ($CoQ_{10}$) and ubiquinol (reduced form) in PBS or ODN treated HBEC3-KT cells were measured using an Agilent 6460 triple quadruple mass spectrometer coupled with an Agilent 1290 series HPLC system. PBS or ODN treated cells were quickly washed with ice-cold PBS and then liquid nitrogen was poured onto cells to rapidly quench metabolic and chemical reactions. To extract ubiquinone and ubiquinol, 100 µl of 100% iso-propanol were added and mixed with the cells. Coenzyme $Q_9$ was added as an internal standard. The cell extracts were vortexed, centrifuged at 17,000 g for 5 minutes at 4°C, and supernatants were transferred to clean auto sampler vials for direct injection. The mobile phase is methanol containing 5 mM ammonium formate. Ubiquinone, ubiquinol and $CoQ_9$ were separated on a Kinetex 2.6 µm C18 100 Å, 100 x 4.6 mm column. The flow rate was 700 µL/min at 37°C. The mass spectrometer was operated in the MRM positive ion electrospray mode with the following transitions: Ubiquinone/oxidized, m/z 880.7→197.1; Uniquinol/reduced, m/z 882.7→197.1; $CoQ_9$ (IS), m/z 795.6 →197.1. Raw files were imported and analyzed using Agilent Mass Hunter Workstation software-Quantitative Analysis.

### Cytochrome $b_H$ measurement

The method of cytochrome $b_H$ measurement was adapted from that described by Quinlan et al. [62]. Isolated mitochondria (1 mg/ml) were resuspended at 37°C in buffer containing 120 mM KCl, 5 mM HEPES, 1 mM EGTA (pH 7.2 at 20°C), and 0.3% (w/v) bovine serum albumin. Absorbance change in Cytochrome $b_H$ was measured after ODN or antimycin A addition using a NanoDrop One UV-Vis spectrophotometer (ThermoFisher Scientific). The Cytochrome $b_H$ signal was recorded by spectrum scanning at the wavelength pair 566–575 nm at 37°C with stirring and normalized based on the assumption that reductions of $b_H$ are 0% with no added substrate and 100% with saturating substrates plus 2 µM DTT. In parallel with cytochrome $b_H$ measurement, mtROS generation was detected in isolated mitochondria (0.5 mg/ml) treated by ODN or antimycin A using mitoSOX red as above described. To determine the effect of the mitochondria membrane potential on Cytochrome $b_H$ redox state and mtROS generation in isolated mitochondria, $b_H^{\%red}$ and mtROS were analyzed after addition of 500 nM FCCP in the assay buffer along with ODN or antimycin A treatment.

### Proteomics analysis

The biotinylated ODN-bound proteins were precipitated with streptavidin beads from mitochondria lysates and resolved by PAGE (Bio-rad). The PAGE gels were stained using a silver

staining kit (Pierce). Silver-stained gel pieces were excised, washed, destained and digested in-gel with 200 ng modified trypsin (sequencing grade, Promega) and Rapigest (TM, Waters Corp.) for 18 h at 37˚C. In-solution samples were precipitated with 5:1 v/v of cold acetone at -20˚C for 18 h, then centrifuged and the acetone was removed prior to treatment with Rapigest (100˚C for 10 min), followed by addition of trypsin. The resulting peptides were extracted and analyzed by high-sensitivity LC-MS/MS on an Orbitrap Fusion mass spectrometer (Thermo Scientific, Waltham MA). Proteins were identified by database searching of the fragment spectra against the SwissProt (EBI) protein database using Mascot (v 2.6, Matrix Science, London, UK) and Proteome Discoverer (v 2.2, Thermo Scientific). Typical search settings were: mass tolerances, 10 ppm precursor, 0.8d fragments; variable modifications, methionine sulfoxide, pyro-glutamate formation; enzyme, trypsin, up to 2 missed cleavages. Peptides were subject to 1% FDR using reverse-database searching.

## RPPA analysis

The reverse phase protein array (RPPA) analyses were performed to examine 161 protein targets in PBS or ODN treated HBEC3-KT cells. Cell lysates were serially diluted in 5 two-fold dilutions with RPPA lysis buffer. An Aushon Biosystems 2470 arrayer (Burlington, MA) was used to print lysates on nitrocellulose-coated FAST slides (Schleicher & Schuell BioScience) and make protein arrays. The array slides were probed with primary antibodies followed by detection with appropriate biotinylated secondary antibodies. The signal was amplified using the Vectastain ABC Elite kit (Vector Laboratories) and visualized by DAB colorimetric reaction. The slides were scanned, analyzed, and quantified using Microvigene software (Vigene-Tech Inc., Carlisle, MA) to generate spot signal intensities, which were processed by the R package SuperCurve (version 1.01). A fitted "supercurve" was plotted with the signal intensities on the Y-axis and the relative log2 concentration of each protein on the X-axis using the non-parametric, monotone increasing B-spline model [87]. Protein concentrations were derived from the supercurve for each lysate by curve-fitting and normalized by median polish [88]. Differential protein expression analysis was performed using R with LIMMA package and adjusted for multiple-testing using the Benjamini-Hoechberg method to reduce the false discovery rate.

## Quantification and statistical analysis

Statistical analyses were performed using SigmaPlot 14.0 (Systat Software, San Jose, CA) and GraphPad Prism 8 (GraphPad Software, San Diego, CA). One-way ANOVA was used to compare the means of multiple treatment conditions or multiple time points. The Holm-Sidak method was used, unless normality testing failed, in which case Kruskall-Wallis method was used. Means of two groups were compared using two-way Student's t-test. Survival comparisons were performed using logrank testing by the Mantel-Cox approach.

## Supporting information

**S1 Table. Candidate mitochondrial peptide targets identified by liquid chromatography-mass spectrometry proteomic analysis.**
(XLSX)

**S2 Table. Regent sources and identifiers.**
(DOCX)

**S3 Table. Experimental models.**
(DOCX)

**S1 Fig. Comparison of mtROS detectors.** (**A**) mtROS production 100 min after HBEC3-KT cells were treated with PBS, Pam2, ODN or Pam2 and ODN as measured by ROSstar fluorescence. (**B**) mtROS production 100 min after HBEC3-KT cells were treated with PBS or ODN as measured by MitoNeoD fluorescence. (**C**) mtROS production 100 min after HBEC3-KT cells were treated with PBS or ODN as measured by MCLA luminescence. * $p \leq 0.001$ vs. PBS by one-way ANOVA using Holm-Sidak method.
(EPS)

**S2 Fig. Comparison of lung epithelial cell types and growth conditions.** mtROS production 100 min after treatment with PBS or ODN was assessed by MitoSOX fluorescence in HBEC3-KT cells plated on collagen (**A**) or without collagen (**B**), in MLE-15 cells plated on collagen (**C**) or without collagen (**D**), in primary mouse tracheal epithelial cells grown submerged on tissue culture plates (**E**) or grown on transwells at air-liquid interface (**F**), in primary mouse alveolar epithelial cells (**G**) and in primary human alveolar epithelial cells (**H**). (**I**) Brightfield (left) and immunofluorescence confocal microscopy (right) from the transwell cultures in **F**. Scale bar = 50 μm. * $p < 0.001$ by Student's t test. CCSP, club cell secretory protein.
(EPS)

**S3 Fig. ODN-induced changes in mitochondrial electron transport chain complex or super-complex activities.** HBEC3-KT cells were exposed to ODN treatment for 0, 50, and 100 min. Electron transfer activities of (**A**) complex I, (**B**) complex I + III, (**C**) rotenone sensitive complex I + III, (**D**) complex II, (**E**) complex II + III, (**F**) complex III, (**G**) complex IV and (**H**) complex V were measured in whole cell lysates. (**I**) Citrate synthase was used as a marker for TCA activity. * $p \leq 0.006$ vs. 0 min ODN by one-way ANOVA using Holm-Sidak method. † $p \leq 0.006$ vs. 0 min ODN by one-way ANOVA using Holm-Sidak method.
(EPS)

**S4 Fig. Uncut immunoblots for mitochondria mass analysis.** HBEC3-KT cells were exposed to ODN for 0, 50, or 100 min. Shown are the uncut immunoblots for the bands shown in Fig 2B. These include (**A**) SDHB, succinate dehydrogenase subunit B; (**B**) COX4, cytochrome c oxidase subunit IV; (**C**) ATP5A, ATP synthase alpha-subunit; (**D**) CS, citrate synthase; (**E**) VDAC1, voltage dependent anion channel 1; and (**F**) β-Actin, used as a loading control.
(EPS)

**S5 Fig. Dosage-dependent effects of ODN on energy metabolite production.** HBEC3-KT cells were treated with PBS or increasing doses of ODN for 50 minutes. Shown are levels of (**A**) ATP, (**B**) ADP, (**C**) AMP, (**D**) NADH and (**E**) NADPH in whole cell lysates. * $p \leq 0.005$ vs. 0 μM ODN by one-way ANOVA using Holm-Sidak method.
(EPS)

**S6 Fig. ODN-induced mtROS production in the setting of TLR signaling deficiency.** mtROS generation as measured by mitoSOX fluorescence 100 min after ODN treatment of primary mouse tracheal epithelial cells from (**A**) wild type vs. *TLR9$^{-/-}$* mice, (**B**) wild type vs. *MyD88$^{-/-}$* mice, (**C**) *MyD88$^{fl/fl}$* vs. *MyD88$^{fl/fl}$;Sftpc-Cre* mice, and (**D**) *Traf6$^{fl/fl}$* vs. *Traf6$^{fl/fl}$;Sftpc-Cre* mice. * $p < 0.005$ vs. PBS in *TLR9$^{-/-}$* or *MyD88$^{-/-}$* cells by one-way ANOVA using Holm-Sidak method. † $p = 0.01$ vs PBS in *MyD88$^{fl/fl}$* cells by Kruskal-Wallis one-way ANOVA using Dunn's method. ‡ $p \leq 0.001$ vs. PBS in *Traf6$^{fl/fl}$* cells by one-way ANOVA using Holm-Sidak method.
(EPS)

**S7 Fig. Biotinylated-ODN precipitation analysis.** Shown are the uncut blots and densitometry from Fig 3E–3G. Human HBEC3-KT cells or murine MLE-15 cells were treated with the indicated biotinylated-ODNs. Mitochondria were isolated and the mitochondrial lysates were incubated with streptavidin beads. The streptavidin precipitants were resolved by polyacrylamide gel electrophoresis, then probed for VDAC1 in (**A**) human or (**B**) mouse mitochondrial lysates following treatment with ODN for the indicated time. Alternately, human mitochondrial lysates were probed for ANT1 (**C**) following treatment with ODN for the indicated time. * p < 0.05 vs. PBS-treated by one-way ANOVA using the Holm-Sidak method. VDAC1, voltage dependent anion channel 1; ANT1, adenine nucleotide translocator 1.
(EPS)

**S8 Fig. Correlation between ODN and VDAC1 pixel intensity by linear regression.** Pixel intensity values for FITC-labeled ODN and Alexa Fluor 555-labeled anti-VDAC1 antibody were plotted against each other and a simple linear regression model was fit to the data. Pearson's correlation coefficients were calculated to determine whether VDAC1 pixel intensity tended to accumulate with ODN1 pixel intensity. FITC, Fluorescein isothiocyanate; VDAC1, voltage dependent anion channel 1.
(EPS)

**S9 Fig. VBIT-4 increases epithelial mtROS generation and $\Delta\Psi_m$.** MitoSOX fluorescence (**A**) and JC-1 ratio (**B**) of HBEC3-KT cells treated for 100 min with PBS, ODN or VDAC1 inhibitor VBIT-4. * p<0.001 by one-way ANOVA.
(EPS)

**S10 Fig. Immunoblots for AMPK-ACC pathway activation analysis.** HBEC3-KT cells were treated with ODN or erastin or cyclosporin A for 0, 50, and 100 min. Cell lysates were resolved by polyacrylamide gel electrophoresis and immunoblotted using antibodies against phospho-AMPK α1, phospho-AMPK α2, phospho-ACC and total AMPK. β-Actin was used as a loading control. (**A**) Uncut blots from Fig 4B and 4C. (**B**) Densitometry studies for the indicated blots relative to β-actin. * p < 0.05 vs PBS-treated by one-way ANOVA using the Holm-Sidak method.
(EPS)

**S11 Fig. Genotyping PCR of _Prkaa1<sup>fl/fl</sup>;Prkaa2<sup>fl/fl</sup>_ mouse tracheal epithelial cells infected with adenovirus containing Cre recombinase or control empty vector.** Shown are post-infection levels of (**A**) _Prkaa1_ and (**B**) _Prkaa2_. AdV, adenovirus.
(EPS)

**S12 Fig. Efficacy of shRNA knock down cells.** Shown are uncut immunoblots for HBEC3-KT transfected with shRNA against (**A**) _CPT1A_, with knockdown #4 (KD4) used in the experiments in Fig 5J; (**B**) _GPD2_, with knockdown #1 (KD1) used in the experiments in Fig 5N and (**C**) _ETFDH_, with knockdown #2 (KD2) used in the experiments in Fig 5O. β-Actin was used as a loading control.
(EPS)

**S13 Fig. Effect of TCA cycle metabolites on ODN-induced mtROS generation.** mtROS dose response to ODN in HBEC3-KT cells supplemented with the TCA metabolites or metabolite analogues (**A**) citrate, (**B**) pyruvate, (**C**) α-ketoglutarate, (**D**) dimethyl succinate, (**E**) dimethyl malonate, (**F**) dimethyl fumarate or (**G**) oxaloacetate. * p≤0.003 vs. 0 μM ODN treated with no metabolite pretreatment by one-way ANOVA using Holm-Sidak method. † p≤0.05 vs. same ODN dose with no metabolite pretreatment by one-way ANOVA using Holm-Sidak method. ‡ p<0.04 vs 0 μM ODN treated with no metabolite pretreatment by Kruskal-Wallis

one-way ANOVA using Dunn's method. § p<0.04 vs 0 μM ODN treated with oxaloacetate pretreatment by Kruskal-Wallis one-way ANOVA using Dunn's method.
(EPS)

**S14 Fig. Alternate electron sources that contribute to mtROS generation.** (**A**) ODN-induced mtROS generation in the presence of absence of glycolysis inhibitor UK5099, β-oxidation inhibitor etomoxir, or glutaminolysis inhibitor BPTES. mtROS dose response to ODN in the presence or absence of (**B**) BPTES, (**C**) etomoxir, or (**D**) UK5099. (**E**) ODN-induced mtROS in the presence of single or combined inhibitors. (**F**) mtROS dose response to ODN in the presence or absence of (2-DG), a D-glucose analogue. (**G**) Mitochondrial oxygen consumption in oligomycin-inhibited HBEC3-KT cells following treatment with ODN in the presence of the indicated inhibitors measured using a Seahorse XFe96 Flux Analyzer, shown as mean ± SEM. 2-DG, 2-deoxy-D-glucose; OCR, oxygen consumption rate. * p≤0.04 vs. no ODN with same pretreatment by one-way ANOVA using Holm-Sidak method. † p≤0.002 vs. same ODN dose with no inhibitor pretreatment by one-way ANOVA using Holm-Sidak method. ‡ p≤0.008 vs. same pretreatment without ODN by one-way ANOVA using Holm-Sidak method. § p< 0.001 vs. ODN treated without inhibitor pretreatment by one-way ANOVA using Holm-Sidak method. Δ p = 0.007 vs PBS treated without inhibitor by Kruskal-Wallis one-way ANOVA using Tukey test. # p < 0.003 vs PBS-treated by one-way ANOVA using Dunnett's test for multiple comparisons.
(EPS)

**S15 Fig. Effect of mitochondrial electron transport chain inhibitors on ODN-induced mtROS generation.** Shown are mtROS dose responses to ODN of HBEC3-KT cells following pre-treatment with (**A**) complex I inhibitor rotenone, (**B**) complex I inhibitor diphenyleneiodonium (DPI), (**C**) complex II inhibitor TTFA, (**D**) complex II inhibitor atpenin, (**E**) complex III inhibitor myxothiazol, (**F**) complex III inhibitor antimycin, (**G**) complex IV inhibitor sodium azide, or (**H**) complex V inhibitor oligomycin. TTFA, 2-thenoyltrifluoroacetone. * p<0.03 vs PBS treated without inhibitor by Kruskal-Wallis one-way ANOVA using Tukey test. † p<0.05 vs same ODN dose without inhibitor by Kruskal-Wallis one-way ANOVA using Tukey test.
(EPS)

**S16 Fig. Scavenging ODN-induced mtROS.** HBEC3-KT cells were treated (or not) with mtROS scavengers mitoTEMPO (**A**) or mitoQ (**B**), then treated with the indicated concentration of ODN, mtROS generation was measured by mitoSOX fluorescence at 100 min after ODN treatment. (**C**) MitoSOX fluorescence of HBEC3-KT cells treated with PBS or ODN prior to inoculation with PBS or *P. aeruginosa*. (**D**) CellROX fluorescence of HBEC3-KT cells treated with PBS or Pam2ODN prior to inoculation with PBS or *P. aeruginosa*. * p≤0.001 vs PBS treated without inhibitor by Kruskal-Wallis one-way ANOVA using Tukey test. † p<0.05 vs same ODN dose without inhibitor by Kruskal-Wallis one-way ANOVA using Tukey test. ‡ p <0.005 vs PBS-treated, uninfected by Kruskal-Wallis one-way ANOVA using Tukey test.
(EPS)

**S17 Fig. Inhibition of ODN-induced increase of mtROS and membrane potential ΔΨm by TTFA and FCCP treatment.** HBEC3-KT cells were pre-treated (or not) with either TTFA or FCCP or both, then treated for 100 min with the indicated concentration of ODN. Shown are mitoSOX fluorescence with (**A**) TTFA, (**B**) FCCP and (**C**) TTFA-FCCP. (**D**) MitoTracker fluorescence, or (**E**) JC-1 ratio of aggregates:monomers with TTFA-FCCP. (**F**) Bacterial burden of HBEC3-KT cells treated with the indicated ligands with or without TTFA or FCCP or both. * p≤0.001 vs PBS treated without inhibitor by Kruskal-Wallis one-way ANOVA using Tukey

test. † p<0.05 vs same ODN dose without inhibitor by Kruskal-Wallis one-way ANOVA using Tukey test. ‡ p<0.001 vs PBS by ANOVA.
(EPS)

## Author Contributions

**Conceptualization:** Yongxing Wang, Lee-Jun Wong, Scott E. Evans.

**Data curation:** Yongxing Wang, Vikram V. Kulkarni, Jezreel Pantaleón García, Miguel M. Leiva-Juárez.

**Formal analysis:** Yongxing Wang, Jezreel Pantaleón García, David L. Goldblatt, Scott E. Evans.

**Funding acquisition:** Scott E. Evans.

**Investigation:** Yongxing Wang, Vikram V. Kulkarni, Miguel M. Leiva-Juárez, David L. Goldblatt, Fahad Gulraiz, Lisandra Vila Ellis, Michael K. Longmire, Sri Ramya Donepudi, Hao Wang.

**Methodology:** Yongxing Wang, Vikram V. Kulkarni, Lisandra Vila Ellis, Jichao Chen, Philip L. Lorenzi.

**Project administration:** Scott E. Evans.

**Resources:** Michael K. Longmire.

**Supervision:** Scott E. Evans.

**Validation:** Yongxing Wang.

**Visualization:** Michael J. Tuvim.

**Writing – original draft:** Yongxing Wang, Scott E. Evans.

**Writing – review & editing:** Yongxing Wang, Jichao Chen, Scott E. Evans.

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
