## [Decision Letter · Decision Letter 0]

15 Mar 2023

Dear Dr. Evans,

Thank you very much for submitting your manuscript "Antimicrobial mitochondrial reactive oxygen species induction by lung epithelial metabolic reprogramming" for consideration at PLOS Pathogens. As with all papers reviewed by the journal, your manuscript was reviewed by members of the editorial board and by several independent reviewers. In light of the reviews (below this email), we would like to invite the resubmission of a significantly-revised version that takes into account the reviewers' comments.

I am returning your manuscript with three reviews. The reviewers came to different conclusions about the paper, as you will see. After reading the reviews and looking at the manuscript, I recommend Major Revision and did not reject your manuscript as recommended by the reviewer #3. Indeed, despite the present form need to address issues raised by reviewers, I have the gut feeling that your study is of interest to tackle antimicrobial resistance by providing new targets. I am sorry I cannot be more positive at the moment, however I am looking forward to receiving your revision. Note that we may send your paper back to some of the more critical reviewers upon resubmission.

I would like to draw your attention to the issue that I have considered critical and for which I expect answers +/- new data. For the other, you are free to decide whether the comments require new data or new explanations arguing in support with your claiming or against the reviewers’ comments.

The first critical issue is your extrapolation to the whole lung. Although, your manuscript represents a large body of work showing that ODN can increase mtROS and thereby being part of the antimicrobial oxidative response of the human airway epithelium, I agree with the reviewer #1 that your results concerned bronchial cells and moreover non-differentiated cells. For these reason, please pay particular attention to the comments #3 and #4 from the reviewer #1.

The second critical issue is the demonstrated link between mtROS production and the alteration of the electron chain transport.

Lines 90-92 (results from figure 1H): inhibiting complex V by oligomycin gives OCR due to proton leak since the ATP-based OCR is inhibited. Therefore, you showed increased OCR with ODN while inhibition by oligomycin. However, would it be possible to measure mtROS production in ODN+oligomycin and (ODN+antimycin or ODN+PBS). It will clearly show increased oxygen consumption by mtROS production.Please address the reviewer #3’s major concerns #2 to confirm how you ruled out any ROS detection other than mtROS that could lead to falsely conclude to mtROS increase.Lines 168-169, you mentioned that AMPK is activated in response to ATP consumption. Thus, when ODN inhibits the ECT and subsequently the ATP production (results 2 and 3), AMPK is activated. Could you clarify why deletions of AMPK genes decreased ODN-induced mtROS?

In addition to the reviewers’ minor comments, I would like you to pay attention to my two minor comments I have listed below.

Line 377, please change “Luria-Bertani medium” for “Lysogeny broth medium” as G. Bertani himself explained that LB has been misconstrued to stand for his name (PMID: 14729683)For the in vivo infection model (lines 370-375) please specify the time of nebulization that is required to ensure a full Pam or AND treatment as well as an effective PA inoculation.

We cannot make any decision about publication until we have seen the revised manuscript and your response to the reviewers' comments. Your revised manuscript is also likely to be sent to reviewers for further evaluation.

Sincerely,

Thomas Guillard, PharmD, PhD

Academic Editor

PLOS Pathogens

David Skurnik

Section Editor

PLOS Pathogens

Kasturi Haldar

Editor-in-Chief

PLOS Pathogens

orcid.org/0000-0001-5065-158X

Michael Malim

Editor-in-Chief

PLOS Pathogens

orcid.org/0000-0002-7699-2064

I am returning your manuscript with three reviews. The reviewers came to different conclusions about the paper, as you will see. After reading the reviews and looking at the manuscript, I recommend Major Revision based on the critiques from the more critical reviews. Indeed, despite the present form need to address issues raised by reviewers, I have the gut feeling that your study is of interest to tackle antimicrobial resistance by providing new targets. I am sorry I cannot be more positive at the moment, however I am looking forward to receiving your revision. Note that we may send your paper back to some of the more critical reviewers upon resubmission.

I would like to draw your attention to the issue that I considered critical and I expect answers +/- new data. For the other, you are free to decide if the comments require new data or new explanation arguing in support with your claiming or against the reviewers’ comments.

The first critical issue is your extrapolation to the whole lung. Although, your manuscript represents a large body of work showing that ODN can increase mtROS and thereby being part of the antimicrobial oxidative response of the human airway epithelium, I agree with the reviewer #1 that concerned only bronchial cells and moreover non-differentiated. For these reason, please pay particular attention to the comments #3 and #4 from the reviewer #1.

The second critical issue is the demonstrated link between mtROS production and the alteration of the electron chain transport.

- Lines 90-92 (results from figure 1H): inhibiting complex V by oligomycin gives OCR due to proton leak since the ATP-based OCR is inhibited. Therefore, you showed increased OCR with ODN while inhibition by oligomycin. However, would it be possible to measure mtROS production in ODN+oligomycin and (ODN+antimycin or ODN+PBS). It will clearly show increased oxygen consumption by mtROS production.

- Please address the reviewer #3’s major concerns #2 to confirm how you ruled out any ROS detection other than mtROS that could lead to falsely conclude to mtROS increase.

- Lines 168-169, you mentioned that AMPK is activated in response to ATP consumption. Thus, when ODN inhibits the ECT and subsequently the ATP production (results 2 and 3), AMPK is activated. Could you clarify why deletions of AMPK genes decreased ODN-induced mtROS?

In addition to the reviewers’ minor comments, I would like you to pay attention to my minor comments I have listed below.

- Line 377, please change “Luria-Bertani medium” for “Lysogeny broth medium” as G. Bertani himself explained that LB has been misconstrued to stand for his name (PMID: 14729683)

- For the in vivo infection model (lines 370-375) please specify the time of nebulization that is required to ensure a full Pam or AND treatment as well as an effective PA inoculation.

Reviewer's Responses to Questions

**Part I - Summary**

Reviewer #1: The report from Yongxing Wang et al. addresses the molecular mechanisms responsible for mitochondrial reactive oxygen species (mtROS) generation after epithelial challenge with TLR ligands. It continues the previous work of the team that developed antimicrobial agents in the context of respiratory research (pneumonia). The investigations are highly significant in the context of antimicrobial strategies for respiratory diseases. The background, the experimental design, and the general execution are robust. The intracellular metabolic cascade leading to ROS production is extremely well characterized by top-notch molecular approaches. The potential involvement of mitochondrial complexes II and III is convincingly described. The figures are appropriate and the results are accurately reported. The methods contained sufficient experimental details. I have only a few remarks that could improve clarity and complement the investigations.

Reviewer #2: Evans and colleagues have developed and functionally characterized a combination of two innate immune stimulators (ODN M362 and Pam2) that induces epithelial resistance against a broad range of opportunistic bacterial and fungal pathogens. In prior work, the authors showed that the induction of mitohondrial ROS is critical to the efficacy of this immunomodulatory strategy that is being tested in human clinical trials. This new manuscript describes the molecular links that bridge Pam2/ODN M362 and mtROS, with the surprising finding that ODN M362 directly binds to the mitochondrial voltage-dependent anion-sensitive channel (VDAC1) in a manner that is independent of TLR9 signaling. This interaction results in metabolic reprograming in mitochondria, and alters cellular nucleotide distribution which in turn activates the AMPK-ACC and fatty acid oxidation. These events promote electron flux through the ETC and the generation of mtROS; ODN M362-dependent disruption of these events results in impaired epithelial mtROS generation and loss of in vivo activity of the ODN M362/Pam 2 activity, resulting in impaired lung host defenses against Pseudomonas challenge in experimental models of disease. Overall, this is an elegant and well-developed study that is relevant for understanding epithelial regulation of innate immunity (and its therapeutic manipulation).

Reviewer #3: This manuscript investigates how ODN induces mitochondrial ROS (mtROS) production via metabolic reprogramming, alters mitochondrial electron transport chain (ETC) activity in a mitochondrial membrane potential (ΔΨm)-dependent manner. They show that antimicrobial ROS are induced from lung epithelial cells by interactions of CpG oligodeoxynucleotides (ODNs) with mitochondrial voltage-dependent anion channel 1

(VDAC1) without dependence on Toll-like receptor 9 (TLR9). The ODN-VDAC1 interaction alters cellular ATP/ADP/AMP localization, increases delivery of electrons to the electron transport chain, enhances mitochondrial membrane potential, and differentially modulates ETC complex activities. These combined effects promote leak of electrons from ETC complex III, resulting in superoxide formation. They conclude that

ODN-induced mitochondrial ROS may yield protective antibacterial effects. Although the studies are interesting and provide some understanding of protective effects of ODN on host immune response, the studies are not enough to conclude that ODN induced mitochondrial ROS generation in epithelial cells is the sole contributor to host protection in vivo in a model of lung infection with P. aeruginosa.

**Part II – Major Issues: Key Experiments Required for Acceptance**

Reviewer #1: (No Response)

Reviewer #2: I do not recommend additional experimentation for acceptance of this manuscript.

Reviewer #3: Major Concerns:

1) The figures that show the seahorse analysis demonstrate effects of oligomycin and antimycin. The entire graph of cellular respiration needs to be shown with basal, maximal OCR, spare respiratory capacity and ATP production. In addition these should be accompanied with graphs and quantification

2) The investigators use both tracheal and bronchial epithelial cells. The rationale for use of these cells needs to be included

3) In Figure 2 F and G they show the expression of NADH and NADPH, however, none of the experiments include quantification of non-mitochondrial ROS. These should be done in combination with infection of bacteria in vitro.

4) Most of the in vitro studies are performed with ODN alone, it will interesting to include experiments with P. aeruginosa and ODN

5) Many of the conclusions are based on use of VDAC1, ANT1, mPTP inhibitors, off target effects of these reagents should be considered

6) The in vivo experiments in Figure 6 show a protective effect of Pam2-ODN on P. aeruginosa infection. From the data and design of experiments it is difficult to conclude that this would be solely due to mitochondrial ROS generation by ODN in epithelial cells in vivo. Multiple cell types particularly phagocytes such as neutrophils, macrophages contribute to host defense in bacterial infections. Although ODN was administered as nebulization, it is likely to have effects on other immune cells.

**Part III – Minor Issues: Editorial and Data Presentation Modifications**

Reviewer #1: 1. The title is somehow misleading for 2 reasons: (i) the work is mainly focused on HBEC3-KT cells that are originating from bronchi, therefore the extrapolation to the entire lung should be tackled with caution; (ii) the level of evidence dealing with the induction of mtROS by the modulators on the short kinetic that is presented hardly demonstrate “metabolic reprogramming”.

2. The production of ROS/mtROS is demonstrated but what about the release of these ROS? In addition, is there a way to test if those ROS are beneficial for the epithelial cells and non-toxic?

3. Some more background and details regarding the origin of the epithelial cells and the related hypothesis that is tested should be given in the text. For example, HBEC3 are mainly used, since they are originating from bronchi they may be ideally suited to analyze the first intra-pulmonary contact of the bacteria but pneumonia particularly affects the alveoli (only MLE-15 was used for only one experimental approach). Is it possible to reproduce the most important findings in another cellular model that would represent small airways or alveoli? It should be at least discussed in the manuscript.

4. The authors should also discuss the limitations of their work regarding the in vitro model that was chosen. HBEC3 are non-differentiated cells, whereas the bacteria first encounter differentiated cells in the airways. In addition, it is not clear why and when cell culture was performed on collagen-coated tissue culture plates and transwells since no data that is presented here seem to discuss influencing parameters such as proliferative state of the cells, level of differentiation, or even general morphology/phenotype. If transwells were used, did the authors try to distinguish non-differentiated vs differentiated airway epithelial cells to test the differential ROS production?

5. The full datasets with all the proteins should be provided in supplemental data regarding Figure 3D and Figure 4A.

Additional comments:

1. The manuscript is generally very well written. Nonetheless, there are a few typos to correct.

2. Is there a label that is not correct in Figure S1A or S1B? As it is I do not understand the difference between those 2 panels.

3. The scale bars on the IF do not appear correct or the cells that are shown would reach a few hundred µm.

4. Is it possible to apply a statistical method to analyze the curves in OCR measurements (F1H, F3L, F4J, F5C, FS12G) and F5D-G?

5. Is it possible to show close inserts for F4D? In addition, it would be interesting to explore the alveoli.

Reviewer #2: There are several minor comments that the authors should address:

1. Figure 2B - please quantify Western blot bands.

2. Line 137-140. Re pulldown assays: Is VDAC1 one of the bands in the 35-50 kDa range - Fig. 3D? Please quantify increase in IP of VDAC1 or ANT1 as a function of ODN exposure.

3. See point 2 for Fig. 4B - please quantify.

4. Figure 4 contains an enormous amount of data and recommend splitting in two figures (panels A - F in one figure; panels G - P in second figure; panel P could be the first panel in the second figure to aid the reader in absorbing the ensuing experiments - currently displayed in panels G - O).

5. Similar considerations for current Fig. 5 - helpful to lead with model - panel N - and then associated data.

Reviewer #3: (No Response)

PLOS authors have the option to publish the peer review history of their article (what does this mean?). If published, this will include your full peer review and any attached files.

Reviewer #1: No

Reviewer #2: **Yes: **Tobias Hohl

Reviewer #3: No
---

## [Decision Letter · Decision Letter 1]

1 Aug 2023

Dear Dr. Evans,

We are pleased to inform you that your manuscript 'Antimicrobial mitochondrial reactive oxygen species induction by lung epithelial immunometabolic modulation' has been provisionally accepted for publication in PLOS Pathogens.

Best regards,

Thomas Guillard, PharmD, PhD

Academic Editor

PLOS Pathogens

David Skurnik

Section Editor

PLOS Pathogens

Kasturi Haldar

Editor-in-Chief

PLOS Pathogens

orcid.org/0000-0001-5065-158X

Michael Malim

Editor-in-Chief

PLOS Pathogens

orcid.org/0000-0002-7699-2064

Reviewer Comments (if any, and for reference):

Reviewer's Responses to Questions

**Part I - Summary**

Reviewer #1: The authors significantly improved their manuscript and satisfactorily answered all my comments. I have no additional remark.

Reviewer #2: The authors have comprehensively responded to my comments and concerns. Well done! It's an elegant study that will be of great interest to the PloS Pathogens readership.

**Part II – Major Issues: Key Experiments Required for Acceptance**

Reviewer #1: NA

Reviewer #2: None.

**Part III – Minor Issues: Editorial and Data Presentation Modifications**

Reviewer #1: NA

Reviewer #2: None.

PLOS authors have the option to publish the peer review history of their article (what does this mean?). If published, this will include your full peer review and any attached files.

Reviewer #1: No

Reviewer #2: **Yes: **Tobias Hohl

---

## [Editor Report · Acceptance letter]

29 Aug 2023

Dear Dr. Evans,

We are delighted to inform you that your manuscript, "Antimicrobial mitochondrial reactive oxygen species induction by lung epithelial immunometabolic modulation," has been formally accepted for publication in PLOS Pathogens.

Best regards,

Kasturi Haldar

Editor-in-Chief

PLOS Pathogens

orcid.org/0000-0001-5065-158X

Michael Malim

Editor-in-Chief

PLOS Pathogens

orcid.org/0000-0002-7699-2064